

# Risk assessment based on a new decision-making approach with fermatean fuzzy sets

Hilal Biderci and Ali F. Guneri

Industrial Engineering, Yildiz Technical University, Istanbul, Turkey

## ABSTRACT

**Background:** This study presents a new approach to decision-making based on the selection of decision-makers according to evaluated criteria in multi-criteria decision-making (MCDM) methods. Therefore, sub-decision-maker groups (SDMGs) are created for each evaluated criterion. The SDMG approach, which is created according to the criteria, offers a more flexible and dynamic structure than the existing approaches. This approach aims to use the expertise and knowledge of decision-makers more effectively. The decision-making approach presented in this study offers an innovative model and adds a new dimension to decision-making processes. This decision-making approach is applied to the plastic injection moulding machine risk assessment, as it involves different criteria. In addition to classical risk parameters such as probability, severity, frequency, and detectability, new parameters such as human error, machine error, and existing safety measures are also used in the risk assessment.

**Methods:** The integration of the analytic hierarchy process (AHP) and the technique for order preference by similarity to ideal solution (TOPSIS) methods into the interval valued fermatean fuzzy set (IVFFS) environment makes an important contribution to a more comprehensive consideration of risks and uncertainties in the risk assessment process. The IVFF-AHP method is used to weight the risk parameters and determine the hazard scores, and the TOPSIS method is used to rank the hazards. A holistic and systematic approach to risk assessment has been achieved by integrating these two methods. Modelling of these methods is carried out using MATLAB_R2024a software.

**Results:** According to the evaluated criteria, it was concluded that the determination of the decision makers separately is applicable to the decision-making process. Identifying the existing safety measures parameter as the most important risk parameter emphasizes the central role of this factor in risk assessment. In addition, machine error and human error parameters are also found to be important in risk assessment. These parameters, which are used for the first time in the literature, offer a broader perspective than traditional methods and provide significant advantages in risk assessment. According to the evaluations, electricity, asphyxiating and toxic gases, and hot water use are determined as the most risky hazards. The sensitivity and comparative analysis performed in the study confirm that the proposed methodology produces consistent and reasonable results.

Corresponding author
Hilal Biderci,
hilal.biderci@std.yildiz.edu.tr

# INTRODUCTION

Decision-making processes are extremely important in situations that require complex and multifaceted decisions, such as multi-criteria decision-making (MCDM) problems. The authorities who directly affect the decision-making process and make the final decisions are the decision-makers. Therefore, decision-makers who carry out the decision-making process are considered important actors in MCDM problems. Decision-makers are also individuals who understand the importance of their decisions and think about how these decisions are made (*Wierzbicki & Wessels, 2000*). Decision-makers' knowledge, skills, priorities, and goals about the problem play a critical role in the success of the decision-making process. Therefore, the correct identification of decision-makers and their participation in decision-making process is a fundamental step in solving MCDM problems. A single decision maker or a group of decision makers typically handles the entire decision-making process in MCDM problems (*Yazdi et al., 2020*; *Fattahi et al., 2020*; *Gul & Ak, 2021*). However, these decision-makers may not have sufficient expertise on every criterion evaluated in the decision-making process. This may result in incorrect or incomplete assessments by decision-makers. Determining decision-makers according to their knowledge of the criteria involved in the decision-making process increases the reliability and information capacity of the decisions made (*Raghunathan, 1999*). In this study, in order to make reliable decisions with high information capacity, decision-makers are divided into sub-decision-maker groups (SDMGs) in accordance with the evaluated criteria. This approach, which is based on identifying decision-makers by dividing them into SDMGs, offers a new approach to the decision-making process. SDMGs are determined to consist of people with appropriate knowledge and expertise on each criterion evaluated in this new decision-making approach. These SDMGs make the evaluations involved in the decision-making processes

This decision-making approach has some unique features. One of them is that different groups and numbers of decision makers can be used for each criterion. Another feature is that a decision maker who has a high weight for one criterion under consideration may have a lower weight for another criterion. Similarly, a decision-maker may be included in the SDMG because of their expertise in a particular criterion but may not be involved in the evaluation of another criterion. The decision-making group may vary for each criterion, as each criterion undergoes separate evaluation. These features emphasise the unique aspect of the study for decision-makers and contribute to the creation of an effective decision-making process by making the right decision. In this study, the proposed new decision-making approach is applied to the risk assessment study based on MCDM methods. Risk assessment is a process in which many different criteria are taken into consideration. Decision-makers can make holistic and sustainable decisions by evaluating all these criteria in the risk assessment. Thus, the correct prioritisation of risks and efficient

use of resources are ensured. However, determining a single decision-maker group for a risk assessment process that includes different criteria may negatively affect the decision results. Therefore, creating SDMGs consisting of experts who have knowledge about the relevant criterion in the evaluation of each criterion may provide more reliable results. For example, different risk parameters such as probability and detectability are evaluated by different decision-makers. As a result, in this study, decision-makers are determined according to the risk parameters to be evaluated. This decision-making approach emphasises the unique aspect of this study in terms of identifying decision-makers.

The risk assessment process includes identifying hazards and determining the value of risk associated with those hazards. Risk parameters are used to determine the value of risk by evaluating different aspects and effects of the risk. The risk parameters to be used in the risk assessment process vary depending on the purpose of the analysis and the subject to be analysed. While fixed risk parameters are used in classical risk assessment methods, the use of different parameters in modern and advanced methods contributes to a better understanding of the different aspects and effects of risk. The use of different risk parameters makes the risk assessment more detailed and comprehensive. In this study, in addition to the probability, severity, frequency and detectability risk parameters used in classical risk assessment methods, new risk parameters such as human error, machine error and existing safety measures are used. These additional risk parameters are intended to assess the contribution of human and machine errors to the value of risk and the effect of existing safety measures on the risk. The main purpose of using these parameters is to examine their direct effects on risk values. The risk parameters of human error, machine error, and existing safety measures are used for the first time in this study in the literature.

In this study, the integration of the AHP method and the TOPSIS method in an interval valued Fermatean fuzzy (IVFF) environment is used for risk assessment. The analytic hierarchy process (AHP) method contributes to the decision-making process by determining the importance level of criteria (*Canco, Kruja & Iancu, 2021*). In this study, the AHP method is used to weight the risk parameters and obtain hazard scores. Then, the technique for order preference by similarity to ideal solution (TOPSIS) method is integrated into the AHP method to prioritise the hazards. Since the TOPSIS method is based on the principle of selecting and ranking the closest to the ideal solution among the alternatives in the decision-making process, it is used to determine the final hazard ranking (*Karuppiah & Sankaranarayanan, 2023*; *Sun, Hu & Liu, 2023*).

The integration of AHP and TOPSIS methods, which are frequently used in decision-making processes, offers significant advantages. In this study, there are several purposes for using AHP and TOPSIS methods. AHP presents decision problems in a hierarchical structure, allowing the decision-making process to be analysed in a systematic way. Other MCDM methods (Best Worst Method: BWM, step-wise weight assessment ratio analysis (SWARA), inter-criteria correlation (CRITIC), full consistency method (FUCOM), *etc.*) do not offer the possibility of such a structured analysis. In addition, thanks to pairwise comparison matrices, it solves the subjective evaluations of decision makers with numerical data. One of the biggest advantages of the

AHP method is that it measures the consistency of decision-makers' assessments. There is no analysis in which consistency control is performed in other MCDM methods. Due to these advantages, the AHP method is frequently preferred over other MCDM methods. In this study, the weighted importance of the criteria and the scores of the alternatives are determined by the AHP method. The importance of the evaluated alternatives according to each criterion is analysed in detail with the AHP method. Since the importance of each alternative is obtained according to each criterion, the alternatives can be compared and interpreted objectively according to these criteria. The criterion weights and scores of the alternatives obtained by the AHP method are transferred directly to the decision matrix. TOPSIS allows the direct use of weights and scores obtained by the AHP method. At this stage, there is no need for a re-evaluation process by decision-makers for the TOPSIS method. In this way, the analysis time is shortened, and the processes are simplified. Due to this feature, integrating the TOPSIS method into the AHP method makes the decision-making processes very practical. Then, through this decision matrix, the TOPSIS method supports the decision process by ranking the alternatives according to their proximity to the ideal solution. With the integration of the AHP-TOPSIS method, clearer and more reliable results are obtained in terms of the measurability of the distances of the alternatives to the positive ideal and negative ideal solutions by creating the decision matrix according to the results obtained with AHP. TOPSIS has a simpler and more straightforward computational structure compared to other MCDM (Višekriterijumsko Kompromisno Rangiranje (VIKOR), Elimination and Choice Expressing Reality (ELECTRE), Organization Method for Enrichment Evaluation (PROMETHEE), *etc.*) methods. As a result, with this integration, the effectiveness of the TOPSIS method increases, and a more transparent, fast and reliable evaluation process is offered to decision-makers. Using these two methods together combines the strengths of both methods, resulting in a more robust and balanced decision-making process. One of the most important advantages of this integration is that the process has a dynamic structure. The weights of the criterion calculated with AHP and the scores of the alternatives may change over time. As a result, rankings may change depending on alternatives. Depending on these changing conditions, results can be easily updated with AHP and TOPSIS. In this way, it is possible to quickly adapt to changing conditions. Due to these advantages, in this study, a comprehensive analysis process for decision-making problems is obtained by combining the ability of the AHP method to determine the importance of criteria and alternatives and the ability of the TOPSIS method to rank alternatives according to their proximity to the ideal solution. As a result, the integration of AHP and TOPSIS methods makes the decision process more systematic, objective and reliable.

Although the integration of AHP and TOPSIS methods offers significant advantages in decision-making processes, real-life decision-making processes often involve uncertainty and hesitation. To better manage this uncertainty, MCDM methods are being extended to fuzzy environments. Thus, uncertainty in decision-making processes is being addressed more effectively (*Cao et al., 2024*). The integration of AHP-TOPSIS methods is insufficient to manage this uncertainty. Therefore, IVFFSs, which have certain advantages among various fuzzy environments, are integrated into these MCDM methods. In this study,

IVFFS is used because of its superiority in managing uncertainty. IVFFSs define membership and non-membership degrees over a wider range, allowing for the expression of uncertainty more precisely. One of the biggest advantages is that decision-makers can model the uncertainty and hesitation in their assessments at a more granular level. This feature enables more robust and reliable decisions to be made, especially in uncertain problems such as risk assessment. It also combines the strengths of fuzzy logic with well-established decision-making techniques, providing a more flexible and realistic evaluation process. Because of these advantages, the MCDM methods used in this study have been extended to the IVFFS environment. Thus, decision-makers can make more informed choices under flexible and complex conditions.

In the application part of this study, a risk assessment is made for the plastic injection moulding machine. The practical aim of this study is to systematically and reliably analyse the risks encountered in plastic injection moulding machines. As a result, one of the methodological aims of this study is to present a new decision-making approach by dividing the comprehensive decision-maker group into SDMGs. It also differs from previous studies in the literature by using new risk parameters such as human error, machine error, and existing safety measures to assess risks. Methodologically, another aim of this study is to integrate and use IVFFAHP and TOPSIS methods in order to obtain more reliable results in an environment of uncertainty. In the risk assessment process, the integration of AHP and TOPSIS methods is used within the framework of IVFFS. These methods are used to determine and rank the value of risks associated with the hazards of the plastic injection moulding machine.

The contributions of this study can be summarised as follows:

- This study includes decision-makers who have an expert level of knowledge in the relevant criterion in the decision-making process, taking into account the specific dynamics of each criterion. The inclusion of domain experts increases the practical acceptability of the results obtained, strengthening the proposed approach in terms of reliability. In particular, with this contribution, this study offers a new methodological framework for the decision-making process.

- This study expands the parameter framework used for risk assessment by evaluating new risk parameters such as human error, machine error, and existing safety measures, as well as traditional risk parameters such as probability, severity, frequency, and detectability. This approach contributes to better decision-making by ensuring that risk assessment is more comprehensive, realistic, and effective.

- In order to demonstrate the applicability of the proposed methodological framework to a real-life problem, an application is made on the assessment of plastic injection moulding machine risks. Due to the gap in the literature, through expert opinions, process analysis and literature review, it identifies the hazards encountered when working with plastic injection moulding machines.

- A hybrid MCDM approach, integrating IVFF-AHP and TOPSIS methods, is presented to assess the risks of the plastic injection moulding machine. This approach combines the importance of various criteria with precise definitions and a more robust alternative

ranking. In addition, by adding flexibility to evaluation processes, this method increases the capacity to adapt to diverse conditions and makes processes more efficient, reliable, dynamic and result oriented. Therefore, the main contribution of the study is that it provides significant support for the creation of a sustainable and comprehensive decision-making process.

- By including IVFFSs, this study provides a detailed and more adaptable means of handling subjective judgements and uncertain data.
- In order to save time for the reusability of the application, it is coded in the MATLAB_R2024a software program and easily analysed for various application areas.

This study consists of the following sections after the introduction: 'Literature Review' presents a comprehensive literature review. 'The Proposed Methodology' details the flowchart of the proposed approach and the methods used. 'Application' presents an application, sensitivity analysis and comparative analysis of the proposed approach. 'Discussion' presents the discussion, and 'Conclusions' presents the conclusion.

## LITERATURE REVIEW

In this section, a comprehensive review study has been conducted to identify the knowledge gaps in the existing literature. Previous studies in the literature on plastic injection moulding machines, risk parameters, Fermatean fuzzy sets (FFSs), the AHP method and the TOPSIS method have been reviewed, and gaps in the literature are identified. In the literature review section, these topics and the motivation for the research are presented.

### Plastic injection moulding machine

Plastic injection moulding machines contain many hazards and risks. Plastic injection moulding machines that are potentially hazardous demand careful handling to ensure OHS. Therefore, it is important for individuals working with injection moulding machines to understand these potential risks and adopt OHS measures. In the literature, studies on this machine generally focus on industrial processes such as supplier and raw material selection, factors affecting plastic production, and evaluation of maintenance strategies (*Duman, Kongar & Gupta, 2019*; *Merizalde-Salas, Zumba-Novay & Peralta-Zurita, 2023*; *Gul, Yucesan & Celik, 2020*; *Pérez-Domínguez et al., 2018*). However, there is a noticeable absence of studies focusing on the assessment of hazards and risks related to OHS through the utilisation of MCDM methods specifically in the context of plastic injection moulding machines.

### Risk parameters in risk assessment

In risk assessment methods, depending on the method used, probability, severity, frequency and detectability parameters are used. When the studies in the literature are examined, a number of different risk parameters have been used other than the risk parameters used in classical risk analysis methods. These risk parameters are undetectability (*Badida, Janakiraman & Jayaprakash, 2023*), cost (*Yazdi et al., 2020*;

*Fattahi et al., 2020*), time (*Yazdi et al., 2020*; *Fattahi et al., 2020*; *Bhattacharjee, Dey & Mandal, 2022*; *Nabizadeh et al., 2021*; *Ghoushchi et al., 2021*), profit (*Yazdi et al., 2020*; *Fattahi et al., 2020*), quality (*Nabizadeh et al., 2021*), operator competency (*Bhattacharjee, Dey & Mandal, 2022*), pollution ratio (*Ghoushchi et al., 2021*), sensitivity to personal protective equipment non-utilisation (*Gul, Ak & Guneri, 2017*; *Liu et al., 2021*), sensitivity to maintenance non-execution (*Gul, Ak & Guneri, 2017*), applicability of preventive measures (*Gul & Ak, 2022*), system reliability (*Yener & Can, 2021*), prevention action opportunity (*Gul & Ak, 2021*), effectiveness (*Gul & Ak, 2021*), maintenance cost (*Valipour et al., 2022*), maintenance duration (*Valipour et al., 2022*) and human error probability (*La Fata et al., 2021*). The use of more risk parameters helps to make more accurate decisions in decision-making processes and to assess risks more effectively. In the literature, it is seen that many different risk parameters are used in addition to the risk parameters used in classical risk assessment methods. This is useful in terms of a better understanding of the different aspects and impacts of risk and the development of an all-round risk assessment strategy.

In this study, the probability, severity, frequency, detectability, human error, machine error and existing safety measure are used as risk parameters. There is not any risk assessment study that includes the parameters of machine error, human error and existing safety measures in the existing literature. The use of these new risk parameters emphasises the unique aspect of this study in the literature.

## Fermatean fuzzy sets

In decision-making processes, MCDM methods and fuzzy logic have been combined and used effectively in solving problems in various fields (*Wan, Dong & Chen, 2024a*, *2024b*). In MCDM methods, fuzzy numbers are used to express the ambiguity of abstract information in a more flexible, understandable and realistic way during evaluations. In this context, FFSs that handle fuzzy logic have gained an important place in the literature and offer the opportunity to model uncertainty more flexibly.

There has been some exploration into their properties, building upon the established understanding of FFSs (*Liu, Liu & Chen, 2019*). Table 1 consolidates information on various studies employing FFSs, along with insights into the application of MCDM methods. The extended literature review is in the File S1. Despite the potential application areas identified for FFSs in existing literature, the utilisation of IVFFSs in comparable domains is yet to be extensively investigated.

When the outputs obtained as a result of the literature review are analysed, it is seen that FFSs have been applied to different MCDM problems. However, the number of studies using IVFF numbers is quite low. According to this literature review, there is no risk assessment study based on the integration of IVFF numbers with the AHP-TOPSIS method. The integration of both methods aims to enhance the approach and provide a more systematic and accurate risk assessment. When the application areas are examined, it is seen that FFSs are used in different areas such as the energy sector, supply chain management, environmental management, construction, logistics and the health sector. For example, numerous studies have been conducted on different topics such as the

**Table 1 Some of the studies on FFSs.**

| Articles | Fuzzy set types | Methods | Application area |
|---|---|---|---|
| Karuppiah, Virmani & Sindhwani (2024) | FF | AHP; DEMATEL | Assessment of Critical Success Factors in the Supply Chain |
| Yalcin & Ayyildiz (2024) | FF | AHP | Prioritizing Vulnerability Factors of Global Food Supply Chains |
| Li & Zhang (2024) | FF | TOPSIS | Evaluate The Data Management Capability of Manufacturing Enterprises |
| Milovanović et al. (2025) | FF | ELECTRE | Ranking of Asset Maintenance Process Key Performance İndicators |
| Yalcin & Ayyildiz (2024) | FF | AHP | Assessment of Food Supply Chain Disruptions |
| Debbarma, Chakraborty & Saha (2024) | FF | SWARA; MABAC | Selection of Healthcare Waste Recycling Technology |
| Li, Gao & Rong (2024) | IVFF | SWARA; EDAS | Selection of Energy Vehicle Battery Supplier |
| Kamali Saraji & Streimikiene (2024) | FF | SWARA; TOPSIS | Assessing Performance in Meeting the Challenges of the Low-Carbon Energy Transition |
| Luo & Liu (2024) | FF | MABAC | The Selection of New Energy Vehicle Power Battery Recycling Service Outlet |
| Wang et al. (2024) | IVFF | CRITIC; PROMETHEE-II | Analysing The Barriers to Resilience Supply Chain Adoption in The Food İndustry |
| Dhruva et al. (2024) | FF | LOPCOW; COCOSO | Selecting Appropriate Cloud Vendors for Healthcare Center |
| Tsai, Shen & Lin (2025) | FF | DEMATEL; TOPSIS | Evaluating Sustainable Development Strategies in the Wire and Cable Industry |
| Ayvaz et al. (2024) | FF | AHP; WASPAS | Occupational Hazards Analysis for Aquaculture Operations |
| Saha et al. (2024) | FF | MULTIMOORA | Evaluating Factors Affecting E-Scooter Selection |
| Aruchsamy et al. (2024) | FF | PROMETHEE | Selection of the Most Suitable Green Supplier for a Construction Company |
| Seikh & Chatterjee (2024) | IVFF | SWARA; BWM, VIKOR | Determining Sustainable Strategies for Electronic Waste Management |
| Rong et al. (2024) | IVFF | LOPCOW; ARAS | Risk Assessment of R&D Projects in Industrial Robot Offline Programming Systems |
| Yu et al. (2024) | FF | COCOSO | Risk Priority of LNG Storage Tank Failure Modes |
| Göçer (2024) | IVFF | ARAS | Prioritizing Renewable Energy Technologies |
| Pamucar et al. (2024) | FF | COCOSO | Green Supplier Selection in Textile Industry |
| Ayyildiz & Erdogan (2024) | FF | SWARA; TOPSIS | Autonomous Vehicle Parking Lot Selection |
| Yildirim et al. (2024) | IVFF | AHP | Assessing the Vulnerability of Urban Road Infrastructure to Seismic Activity |
| Gao et al. (2024) | FF | BWM; VIKOR | Selecting Health Care Waste Treatment Technology |
| Seikh & Chatterjee (2024) | IVFF | SWARA; ARAS | Determination of Best Renewable Energy Sources |
| Golui, Mahapatra & Mahapatra (2024) | FF | TOPSIS | Electric Vehicle Selection |
| Kirişci et al. (2025) | IVFF | WASPAS | Green Supply Chain Management |
| Majd et al. (2025) | FF | TOPSIS | Ranking the Strategies Based on Business Intelligence in the Context of Smart City |
| Dharmalingam, Mahapatra & Vijayakumar (2025) | FF | TOPSIS | Selecting a Site for a Mobile Tower Installation |
| Yildirim & Ayyildiz (2025) | FF | BWM; WASPAS | Selecting 3D Printing Technology |

| Table 1 (continued) | | | |
|---|---|---|---|
| **Articles** | **Fuzzy set types** | **Methods** | **Application area** |
| *Yazar Okur et al. (2025)* | FF | WASPAS | Evaluating Logistics Sector Sustainability Indicators |
| Current Study | IVFF | AHP; TOPSIS | Assessment of Plastic Injection Moulding Machine Risks |

**Note:**
Fermatean Fuzzy (FF); Interval Valued Fermatean Fuzzy (IVFF); Technique for Order Performance by Similarity to Ideal Solution (TOPSIS); Multi-Objective Optimization on the basis of Ratio Analysis (MULTIMOORA); Weighted Aggregated Sum Product Assessment (WASPAS); Criteria Importance Through Intercriteria Correlation (CRITIC); Step-wise Weight Assessment Ratio Analysis (SWARA); Evaluation based on Distance from Average Solution (EDAS); Multi-Attributive Border Approximation area Comparison (MABAC); Decision Making Trial and Evaluation Laboratory (DEMATEL); Analytic Hierarchy Process (AHP); Elimination and Choice Expressing Reality (ELECTRE); Combined Compromise Solution (CoCoSo); VIšekriterijumsko KOmpromisno Rangiranje (VIKOR); Preference Ranking Organization Method for Enrichment Evaluation (PROMETHEE); Logarithmic percentage change-driven objective weighting (LOPCOW); Best-Worst Method (BWM); Additive Ratio Assessment (ARAS).

evaluation of renewable energy sources (*Göçer, 2024*; *Seikh & Chatterjee, 2024*), the selection of electric vehicles and charging stations (*Golui, Mahapatra & Mahapatra, 2024*), and waste management (*Debbarma, Chakraborty & Saha, 2024*; *Seikh & Chatterjee, 2024*).

As a result, this study aims to improve the decision-making process using AHP and TOPSIS methods based on the field of plastic injection moulding machine risk assessment.

## The AHP method in risk assessment methods

The AHP method is one of the most widely used MCDM methods for prioritising criteria (*Saaty, 1994*). This method offers the opportunity to evaluate both qualitative and quantitative criteria together by including the priorities of the decision maker in the decision process (*Ayyildiz & Taskin Gumus, 2020*). It simplifies complex decision-making problems by creating a hierarchical structure (*Gul, 2018*). In addition, decision-makers make a pairwise comparison of criteria in the AHP method and determine the importance of these criteria (*Bakioglu & Atahan, 2021*; *Biswas et al., 2025*). One of the most important advantages of using the AHP method is that it allows the consistency of the decision-maker's evaluations to be measured, and the group has a structure suitable for decision-making. AHP and AHP's fuzzy versions are widely used decision-making methods, especially in risk assessment processes (*Gul, 2018*). They offer an objective and systematic assessment process. The AHP and AHP's fuzzy extensions used in risk assessment studies in the literature are widely used in the weighting of risk parameters (*Gul & Guneri, 2016*; *Ayvaz et al., 2024*) and in the assessment of hazards (*Aminbakhsh, Gunduz & Sonmez, 2013*; *Othman et al., 2016*). In this study, the AHP method is used to determine the weight of risk parameters and to obtain hazard scores.

## The TOPSIS method in risk assessment methods

TOPSIS is one of the MCDM methods. This method is used effectively, especially in solving complex decision problems (*Biscaia, Junior & Colmenero, 2021*; *Hwang & Yoon, 1981*). The TOPSIS method calculates the distances of alternatives from positive ideal and negative ideal solutions and offers the opportunity to rank them according to their proximity to the best solution (*Hezer, Gelmez & Özceylan, 2021*; *Kizielewicz & Sałabun, 2024*). Thanks to its distance-based evaluation capability, it provides an effective approach in cases where criteria are not of equal importance and delicate balances need to be

observed. It also has the advantage of being able to process a large number of alternatives and criteria without affecting the calculation time (*Jati, 2012*). The TOPSIS method and its fuzzy extensions are widely used in risk assessment processes. The TOPSIS method has found wide application in the field of risk assessment in the literature (*Ali & Maryam, 2014*; *Jozi & Majd, 2014*). In this study, to rank the hazards of plastic injection moulding machines and to determine the risk coefficient value, the TOPSIS method is used.

### Research motivation

Although current decision-making methods offer significant advantages in evaluating different criteria and alternatives, there is a need for approaches that can systematically and flexibly integrate the views of decision-makers in complex problems. The main motivation of this study is to develop a new decision-making framework for the separation of an inclusive decision-making group into SDMGs according to evaluation criteria. This new decision-making framework is applied to the risk assessment process, where many different criteria are evaluated. Due to the gap in the literature, the risk assessment of the hazards of the plastic injection moulding machine is carried out. New risk parameters are used in this risk assessment process to go beyond traditional approaches to risk parameters and make the risk assessment more detailed and comprehensive. This allows a multidimensional assessment of the impact of risk parameters on risk value and offers a new perspective. This study presents a method based on the integration of AHP and TOPSIS methods under IVFFS for risk assessment. AHP is preferred in this study because it provides a systematic analysis of the decision-making process, uses pairwise comparison matrices and provides consistency control of these matrices. TOPSIS is preferred because of the practicality and ease of application in transferring the weights and scores obtained by the AHP method directly in the decision matrix. With the integration of these two methods, it is ensured that the analysis process is shortened and has a dynamic structure. This study is extended to the IVFFS environment due to the ability to model the uncertainty and hesitation in decision-makers' evaluations at a more granular level.

## THE PROPOSED METHODOLOGY

The proposed approach is divided into three basic phases. In the study, first, the risk parameters, hazards and decision-makers are determined. A hierarchical framework of criteria and alternatives is created. Then, the implementation phases of the proposed approach are started. In the first phase, the IVFF-AHP method is used to determine the weights of the risk parameters. In the first step of IVFF-AHP, expert opinions are collected, and a pairwise comparison matrix is created using the linguistic terms given in Table 2. Then, the procedures are performed according to the procedures between Steps 1 and 10 as described in the steps of the IVFFAHP-TOPSIS method given in this section. At the end of step 10, the normalised weights of each risk parameter are obtained. In the second phase, the IVFF-AHP method is applied to evaluate the hazards according to the risk parameters. The pairwise comparison matrices of the hazards are created according to each risk parameter. The hazards are then assessed by the decision-makers according to the linguistic terms in Table 2. Then the procedures are performed according to the

**Table 2 Linguistic terms and IVFF number equivalents.**

| Linguistic terms | IVFF number equivalent | | | |
|---|---|---|---|---|
| | $\mu_L$ | $\mu_U$ | $v_L$ | $v_U$ |
| Certainly High Importance (CHI) | 0.95 | 1 | 0 | 0 |
| Very High Importance (VHI) | 0.8 | 0.9 | 0.1 | 0.2 |
| High Importance (HI) | 0.7 | 0.8 | 0.2 | 0.3 |
| Slightly More Importance (SMI) | 0.6 | 0.65 | 0.35 | 0.4 |
| Equally Importance (EI) | 0.5 | 0.5 | 0.5 | 0.5 |
| Slightly Less Importance (SLI) | 0.35 | 0.4 | 0.6 | 0.65 |
| Low Importance (LI) | 0.2 | 0.3 | 0.7 | 0.8 |
| Very Low Importance (VLI) | 0.1 | 0.2 | 0.8 | 0.9 |
| Certainly Low Importance (CLI) | 0 | 0 | 0.95 | 1 |

procedures between Steps 1 and 10 as described in the steps of the IVFFAHP-TOPSIS method given in this section. At the end of this stage, scores of hazards are obtained according to each risk parameter. After determining the risk parameter weights and hazard scores, the TOPSIS method is applied to determine the risk coefficient value. The risk parameter weights obtained in Phase 1 and the scores of the hazards obtained from Phase 2 are transferred directly to the decision matrix. The rest of the steps are performed according to the procedures between Steps 11–15 as described in the section of the IVFFAHP-TOPSIS method. The ranking of each hazard is obtained at the end of step 15. In the proposed approach, the integration of IVFFAHP and TOPSIS creates a robust decision-making framework that considers both the importance of the criteria and alternatives and the relative performance of the alternatives. Finally, the accuracy of the evaluation results is tested with sensitivity analysis, and the reliability of the analysis results is increased by comparing with different methods. The phases and steps of the proposed approach are presented in Fig. 1. The coding of these methods is done in the MATLAB_R2024a software program to model IVFFAHP-TOPSIS. The codes for the MATLAB software program are given from Files S2 to S10. The proposed approach is further detailed in the following subsections.

## Preliminaries on IVFFSs

Some preliminary information about IVFFs and related notations is provided in this subsection. FFSs are an extension of orthopair fuzzy clusters that can more easily process ambiguous information in the decision-making process (*Garg, Shahzadi & Akram, 2020*). The concept of FFS was developed by *Senapati & Yager (2020, 2019)* as a generalisation of intuitive fuzzy sets (IFSs) and Pythagorean fuzzy sets (PFSs). In IFSs, the sum of the membership and non-membership degrees should be at most one, and in PFSs, the sum of their squares should be at most one (*Senapati & Yager, 2020*). To offer more flexibility to decision-makers, FFS has been developed such that the sum of the cubes of the membership and non-membership degrees is equal to 1 (see Fig. 2). In decision-making processes, decision-makers are confronted with hesitation and partial information that

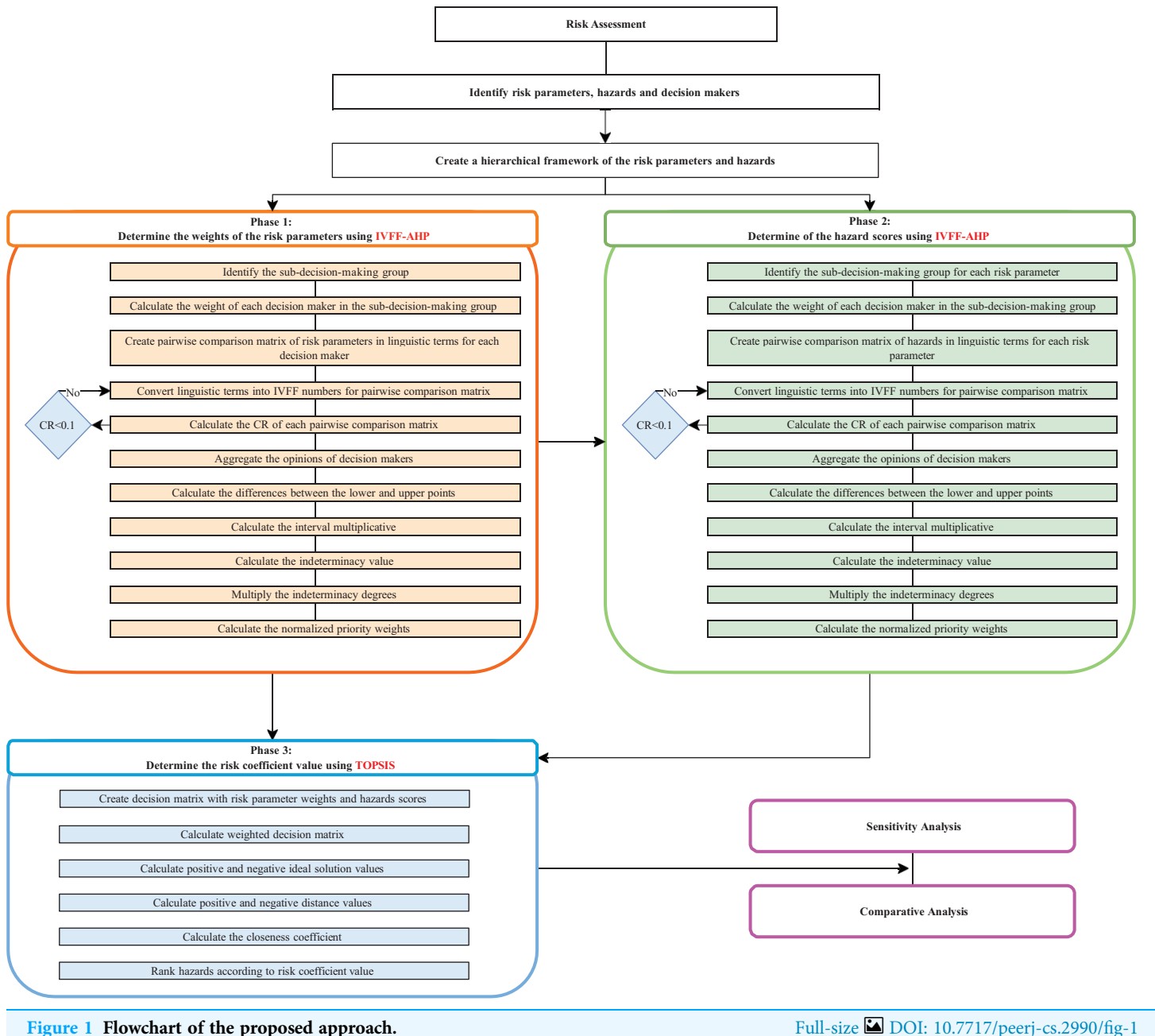

**Figure 1 Flowchart of the proposed approach.**

other fuzzy numbers cannot fully express (*Simić et al., 2022*). FFSs evaluated the uncertainty more comprehensively, taking into account the hesitations and partial information that decision-makers cannot fully articulate. This feature makes FFSs especially effective in situations such as risk assessment, in which there are opinions of decision-makers and uncertain data. FFSs increase credibility in the decision-making process by reflecting expert opinions more accurately. FFSs are more convenient and better suited than other indeterminate cluster extensions to address uncertainty by assigning membership and non-membership rank parameters to a larger domain. This set can be

**Peer**J Computer Science

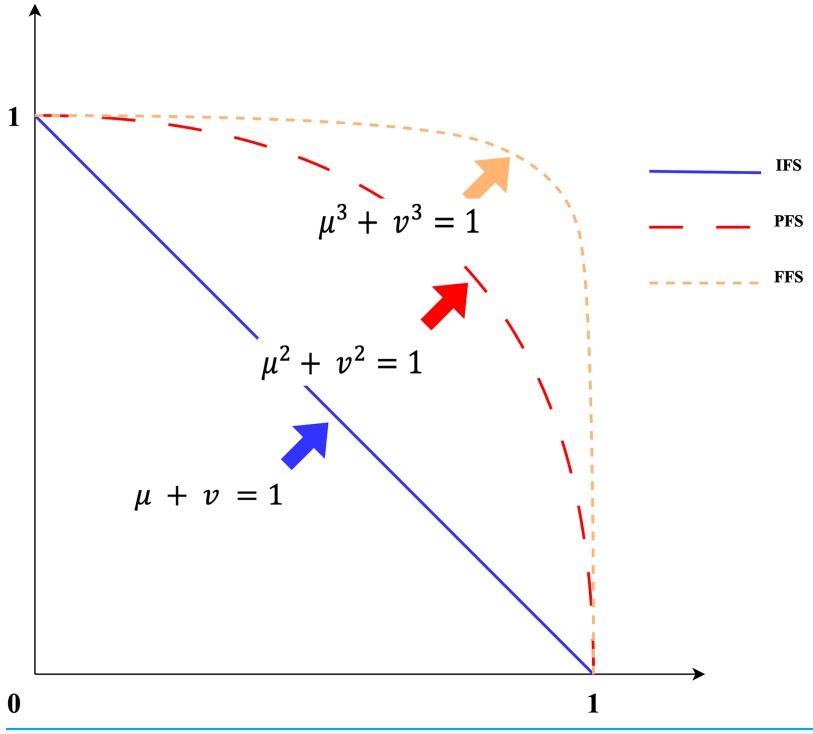

**Figure 2 Comparison IFSs, PFSs and FFSs.**

described as an innovative approach to represent unreliable, imprecise, and uncertain information in a fuzzy environment (*Garg, Shahzadi & Akram, 2020*; *Simić et al., 2022*). As a generalisation of FFSs, IVFFs are used in this study because they allow flexible comparison of range-valued numerical data. For example, suppose a decision maker defines the membership value as [0.65, 0.7] non-membership value as [0.60, 0.75] of an alternative. Here, the sum of the squares of the upper bound of the membership value $(0.7)^2$ and the upper bound of the non-membership value $(0.75)^2$ is greater than 1. In this case, it is neither IVIFS nor IVPFS. However, $(0.7)^3 + (0.75)^3 = 0.7648 \leq 1$. In this case, it can be considered as IVFFS.

Linguistic terms are used by decision-makers in the assessment of criteria and hazards. These linguistic terms utilise fuzzy numbers to express the uncertainty of abstract information in a more flexible, understandable, and realistic way in the evaluation processes conducted by decision makers. For this reason, the linguistic terms and their corresponding IVFF number equivalents are given in Table 2 (*Jeevaraj, 2021*).

Operations for IVFFs are given below (*Jeevaraj, 2021*):

**Definition 1.** If we define the universe of discourse as X, the fermatean fuzzy number in this set is presented as in Eq. (1) (*Jeevaraj, 2021*).

$$\tilde{F} = \{x, \mu_{\tilde{F}}(x), v_{\tilde{F}}(x); x \in X\} \tag{1}$$

where $\mu_{\tilde{F}}(x) \mapsto [0, 1]$ and $v_{\tilde{F}}(x) \mapsto [0, 1]$ shows the closed intervals of the membership and non-membership degrees of the element $x \in X$ to the set $\tilde{F}$, respectively. Also, the lower

and upper limits are denoted by $\mu_{\tilde{F}}^L(x)$, $\mu_{\tilde{F}}^U(x)$, $v_{\tilde{F}}^L(x)$, $v_{\tilde{F}}^U(x)$, respectively. Therefore, $\tilde{F}$ can also be expressed as in Eqs. (2) and (3).

$$\mu_{\tilde{F}}(x) = \left[\, \mu_{\tilde{F}}^L(x),\ \mu_{\tilde{F}}^U(x)\,\right] \mapsto [0,1] \tag{2}$$

$$v_{\tilde{F}}(x) = \left[v_{\tilde{F}}^L(x),\ v_{\tilde{F}}^U(x)\right] \mapsto [0,1] \tag{3}$$

$$0 \le \mu_{\tilde{F}}^U(x)^3 + v_{\tilde{F}}^U(x)^3 \le 1. \tag{4}$$

For each $x \in X$, $\pi_{\tilde{F}}(x) = \left[\pi_{\mathcal{F}}^L(x), \pi_{\mathcal{F}}^U(x)\right]$ is called as the hesitancy degree in IVFFSs. Lower and upper limits of the hesitancy degree:

$$\pi_{\tilde{F}}^L(x) = \sqrt[3]{1 - \left(\mu_{\tilde{F}}^U(x))^3 + (v_{\tilde{F}}^U(x)\right)^3} \tag{5}$$

$$\pi_{\tilde{F}}^U(x) = \sqrt[3]{1 - \left(\mu_{\tilde{F}}^L(x))^3 + (v_{\tilde{F}}^L(x)\right)^3}. \tag{6}$$

**Definition 2.** $\tilde{F} = \left(\left[\mu_{\tilde{F}}^L(x),\ \mu_{\tilde{F}}^U(x)\right], \left[v_{\tilde{F}}^L(x),\ v_{\tilde{F}}^U(x)\right]\right)$, $\tilde{F}_1 = \left(\left[\mu_{\tilde{F}1}^L(x),\ \mu_{\tilde{F}1}^U(x)\right], \left[v_{\tilde{F}1}^L(x),\ v_{\tilde{F}1}^U(x)\right]\right)$ and $\tilde{F}_2 = \left(\left[\mu_{\tilde{F}2}^L(x),\ \mu_{\tilde{F}2}^U(x)\right], \left[v_{\tilde{F}2}^L(x),\ v_{\tilde{F}2}^U(x)\right]\right)$ be three IVFFSs $\lambda > 0$, their operations are as follows (*Jeevaraj, 2021*):

$$\tilde{\mathcal{F}}_1 \oplus \tilde{\mathcal{F}}_2 = \left(\begin{bmatrix} \sqrt[3]{\left(\mu_{\tilde{\mathcal{F}}_1}^L\right)^3 + \left(\mu_{\tilde{\mathcal{F}}_2}^L\right)^3 - \left(\mu_{\tilde{\mathcal{F}}_1}^L\right)^3\left(\mu_{\tilde{\mathcal{F}}_2}^L\right)^3,} \\ \sqrt[3]{\left(\mu_{\tilde{\mathcal{F}}_1}^U\right)^3 + \left(\mu_{\tilde{\mathcal{F}}_2}^U\right)^3 - \left(\mu_{\tilde{\mathcal{F}}_1}^U\right)^3\left(\mu_{\tilde{\mathcal{F}}_2}^U\right)^3} \end{bmatrix}, \left[v_{\tilde{\mathcal{F}}_1}^L v_{\tilde{\mathcal{F}}_2}^L, v_{\tilde{\mathcal{F}}_1}^U v_{\tilde{\mathcal{F}}_2}^U\right]\right) \tag{7}$$

$$\tilde{\mathcal{F}}_1 \otimes \tilde{\mathcal{F}}_2 = \left(\left[\mu_{\tilde{\mathcal{F}}_1}^L \mu_{\tilde{\mathcal{F}}_2}^L, \mu_{\tilde{\mathcal{F}}_1}^U \mu_{\tilde{\mathcal{F}}_2}^U\right], \begin{bmatrix} \sqrt[3]{\left(v_{\tilde{\mathcal{F}}_1}^L\right)^3 + \left(v_{\tilde{\mathcal{F}}_2}^L\right)^3 - \left(v_{\tilde{\mathcal{F}}_1}^L\right)^3\left(v_{\tilde{\mathcal{F}}_2}^L\right)^3} \\ \sqrt[3]{\left(v_{\tilde{\mathcal{F}}_1}^U\right)^3 + \left(v_{\tilde{\mathcal{F}}_2}^U\right)^3 - \left(v_{\tilde{\mathcal{F}}_1}^U\right)^3\left(v_{\tilde{\mathcal{F}}_2}^U\right)^3} \end{bmatrix}\right) \tag{8}$$

$$\lambda\tilde{\mathcal{F}} = \left(\left[\sqrt[3]{1 - \left(1 - \left(\mu_{\tilde{\mathcal{F}}}^L\right)^3\right)^\lambda}, \sqrt[3]{1 - \left(1 - \left(\mu_{\tilde{\mathcal{F}}}^U\right)^3\right)^\lambda}\right], \left[\left(v_{\tilde{\mathcal{F}}}^L\right)^\lambda, \left(v_{\tilde{\mathcal{F}}}^U\right)^\lambda\right]\right) \tag{9}$$

$$\tilde{\mathcal{F}}^\lambda = \left(\left[\left(\mu_{\tilde{\mathcal{F}}}^L\right)^\lambda, \left(\mu_{\tilde{\mathcal{F}}}^U\right)^\lambda\right], \left[\sqrt[3]{1 - \left(1 - \left(v_{\tilde{\mathcal{F}}}^L\right)^3\right)^\lambda}, \sqrt[3]{1 - \left(1 - \left(v_{\tilde{\mathcal{F}}}^U\right)^3\right)^\lambda}\right]\right). \tag{10}$$

**Definition 3.** Let $\tilde{F}_i = \left(\left[\mu_{\tilde{\mathcal{F}}_i}^L(x), \mu_{\tilde{\mathcal{F}}_i}^U(x)\right], \left[v_{\tilde{\mathcal{F}}_i}^L(x), v_{\tilde{\mathcal{F}}_i}^U(x)\right]\right)$ ($i = 1, 2, \ldots, n$) be a set of IVFFSs and $w = (w_1, w_2, \ldots, w_n)^T$ be a weight vector of $\mathcal{F}_i$ with $\sum_{i=1}^n w_i = 1$ (*Jeevaraj, 2021*),

Interval-Valued Fermatean Fuzzy Weighted Average (IVFFWA) operator is a mapping IVFFWA:

$$\tilde{\mathcal{F}}^n \to \tilde{\mathcal{F}}, \text{ where IVFFWA } (\tilde{\mathcal{F}}_1, \tilde{\mathcal{F}}_2, \ldots, \tilde{\mathcal{F}}_n)$$

$$
= \left( \left[ \sqrt[3]{\left( 1 - \prod_{i=1}^{n}(1 - (\mu_{\tilde{\mathcal{F}}_i}^L)^3)^{w_i} \right)}, \sqrt[3]{\left( 1 - \prod_{i=1}^{n}(1 - (\mu_{\tilde{\mathcal{F}}_i}^U)^3)^{w_i} \right)} \right] \right.
$$

$$
\left. \times \left[ \prod_{i=1}^{n}\left( v_{\tilde{\mathcal{F}}_i}^L \right)^{w_i}, \prod_{i=1}^{n}(v_{\tilde{\mathcal{F}}_i}^U)^{w_i} \right] \right). \tag{11}
$$

Interval-valued fermatean fuzzy weighted geometric (IVFFWG) operator is a mapping IVFFWA: $\tilde{\mathcal{F}}^n \to \tilde{\mathcal{F}}$, where $IVFFWA\,(\tilde{\mathcal{F}}_1, \tilde{\mathcal{F}}_2, \dots, \tilde{\mathcal{F}}_n)$

$$
= \left( \left[ \prod_{i=1}^{n}\left( \mu_i^L \right)^{w_i}, \prod_{i=1}^{n}\left( \mu_i^U \right)^{w_i} \right] \right.
$$

$$
\left. \times \left[ 3\left( 1 - \prod_{i=1}^{n}\left( 1 - \left( v_{\tilde{\mathcal{F}}_i}^L \right)^3 \right)^{w_i} \right), \sqrt[3]{\left( 1 - \prod_{i=1}^{n}\left( 1 - \left( v_{\tilde{\mathcal{F}}_i}^U \right)^3 \right)^{w_i} \right)} \right] \right). \tag{12}
$$

### IVFFAHP-TOPSIS method

This section presents the main steps and equations of the IVFFAHP-TOPSIS integrated method. The IVFFAHP method was developed by *Alkan & Kahraman (2023)*, and this study follows the steps suggested for IVFFAHP. The equations of the IVFFAHP method are given from Step 1 to Step 10. Then, in this study, the classical TOPSIS method is integrated into the IVFFAHP method. The equations of the TOPSIS method are given from Step 11 to Step 15.

Step 1. Create the hierarchical framework by determining the criteria and alternatives. The set $A_i = (A_1, A_2, \dots, A_n)$, having i $= (1, 2, \dots, n)$ alternatives, is evaluated by $m$ criteria of set $C_j = (C_1, C_2, \dots, C_m)$, with j $= (1, 2, \dots, m)$. Let $w_j = (w_1, w_2, \dots, w_m)$ be the vector set represented the criteria weights, where $w_j > 0$ and $\sum_{j=1}^{m} w_j = 1$.

Step 2. Decision makers are evaluated and the weight for each decision maker is determined. Table 3 is used to evaluate decision makers. At the same time, the weight of $t$ experts are denoted by $\psi^t$, where $k$ is the number of experts. The sum of all expert weights is $\sum_{t=1}^{k} \psi^t = 1$. Table 2 presents linguistic terms and their corresponding IVFFSs. Let d be the number of decision makers, and $DM_t = (\mu_t, v_t)$ be the corresponding FFSs to determine weight of decision maker t based on evaluation. Each decision maker is evaluated individually, and the weight of decision maker is determined according to Eq. (13).

$$
\psi_t = \frac{S^+(DM_t)}{\sum_{t=1}^{d} S^+(DM_t)} = \frac{1 + \mu_t^3 - v_t^3}{\sum_{t=1}^{d} (1 + \mu_t^3 - v_t^3)}. \tag{13}
$$

Step 3. Created the pairwise comparison matrix $Z = (z_{ij})_{m \times m}$ based on the linguistic variables of decision makers given in Table 2.

$$
Z = \begin{bmatrix} 1 & z_{12} & \cdots & z_{1m} \\ z_{21} & 1 & \cdots & z_{2m} \\ \vdots & \vdots & \ddots & \vdots \\ z_{m1} & z_{m2} & \cdots & 1 \end{bmatrix}, \text{ where } z_{ij} = \left\langle \left[ \mu_{ij}^L, \mu_{ij}^U \right], \left[ v_{ij}^L, v_{ij}^U \right] \right\rangle. \tag{14}
$$

**Table 3 Linguistic terms and FF number equivalents.**

| Linguistic terms | μ | v |
|---|---|---|
| Absolutely Skilled-AS | 0.95 | 0.10 |
| Very Skilled-VS | 0.75 | 0.30 |
| More Skilled-MS | 0.55 | 0.50 |
| Skilled-S | 0.30 | 0.75 |
| Less Skilled-LS | 0.10 | 0.95 |

Step 4. Calculated for the consistency ratio (CR) of each pairwise comparison matrix (Z). The CR of the matrix is determined according to Satty's classical consistency procedure. The random index (RI) is given in File S11. $\lambda\_max$ represents the highest eigenvalue, while $n$ represents the number of criteria.

$$CR = \frac{CI \text{ (Consistency Index)}}{RI} \tag{15}$$

$$CI = \frac{\lambda\_\max - n}{n - 1}. \tag{16}$$

This step ensures the logical coherence and internal consistency of the decision-maker judgments. By applying Saaty's CR, the reliability of the pairwise comparison matrices are validated. This step is essential to minimize subjective inconsistency in the decision-making process.

Step 5. Aggregate the opinions of decision makers. The pairwise comparison matrix constituted for each decision maker is aggregated by using IVFFWA aggregation operator given in Eq. (17). Let $A_{ij}^t = \left( \left[ \mu_{ij}^{Lt}, \mu_{ij}^{Ut} \right], \left[ v_{ij}^{Lt}, v_{ij}^{Ut} \right] \right)$ be the pairwise comparison of criteria $i$ and $j$ by decision maker t. IVFFWA $(\tilde{\mathcal{F}}_1, \tilde{\mathcal{F}}_2, \ldots, \tilde{\mathcal{F}}_n)$

$$= \left( \left[ \sqrt[3]{\left( 1 - \prod_{i=1}^{n} \left( 1 - \left( \mu_{\tilde{\mathcal{F}}_i}^L \right)^3 \right)^{w_i} \right)}, \sqrt[3]{\left( 1 - \prod_{i=1}^{n} \left( 1 - \left( \mu_{\tilde{\mathcal{F}}_i}^U \right)^3 \right)^{w_i} \right)} \right] \times \left[ \prod_{i=1}^{n} \left( v_{\tilde{\mathcal{F}}_i}^L \right)^{w_i}, \prod_{i=1}^{n} \left( v_{\tilde{\mathcal{F}}_i}^U \right)^{w_i} \right] \right). \tag{17}$$

Step 6. Find the differences matrix $D = (d_{ij})_{m \times m}$ between lower and upper points of the membership and non-membership functions using Eqs. (18) and (19).

$$d_{ij}^L = \left( \mu_{ij}^L \right)^3 - \left( v_{ij}^U \right)^3 \tag{18}$$

$$d_{ij}^U = \left( \mu_{ij}^U \right)^3 - \left( v_{ij}^L \right)^3. \tag{19}$$

In this step, the differences between membership and non-membership degrees are calculated and the superiority and uncertainty levels of the criteria compared to each other are quantitatively revealed. The differences obtained form the basis for a more accurate calculation of the criterion importance in the next steps.

Step 7. Find the interval multiplicative matrix $S = \left(s_{ij}\right)_{m \times m}$ using Eqs. (20) and (21).

$$s_{ij}^L = \sqrt[3]{1000^{d_{ij}^L}} \tag{20}$$

$$s_{ij}^U = \sqrt[3]{1000^{d_{ij}^U}}. \tag{21}$$

In this step, exponential transformation is applied to increase or decrease the effect of differences. Thanks to this transformation, small differences are suppressed, while the effect of large differences is highlighted. This makes it easier to identify critical criteria in the decision-making process.

Step 8. Obtain the indeterminacy value $T = \left(t_{ij}\right)_{m \times m}$ of the $z_{ij}$ is Eq. (22).

$$t_{ij} = 1 - \left(\mu_{ij_U}^3 - \mu_{ij_L}^3\right) - \left(v_{ij_U}^3 - v_{ij_L}^3\right). \tag{22}$$

This step reflects the degree of indeterminacy between assessments. Uncertainty in an FF environment refers to the level of indecision in decision-makers' assessments, and the inclusion of these indeterminacies in weight calculations increases the reliability of evaluations.

Step 9. Multiply the indeterminacy degrees with $S = \left(s_{ij}\right)_{m \times m}$ matrix to find the matrix of unnormalized weights $R = \left(r_{ij}\right)_{m \times m}$ using Eq. (23).

$$r_{ij} = \left(\frac{s_{ij}^L + s_{ij}^U}{2}\right) t_{ij}. \tag{23}$$

In this step, criterion weights are determined by considering both the relative differences between the criteria and the indeterminacy levels of the evaluations. Thus, both the importance and the reliability element were included in the evaluation at the same time.

Step 10. Obtain the normalised priority weights $w_i$ by using Eq. (24).

$$w_i = \frac{\sum_{j=1}^m r_{ij}}{\sum_{i=1}^m \sum_{j=1}^m r_{ij}}. \tag{24}$$

This step allows the sum of the weights of all criteria to be normalized to 1. Thus, decision-makers can clearly and comparably see the relative importance of each criterion in the overall decision process.

Step 11. A weighted decision matrix $V = \left(v_{ij}\right)_{n \times m}$ is created using the criteria and alternatives. Where $w_j$ is the criterion weight. $a_{ij}$ is the evaluation of the i. alternative according to the $j$. criterion.

$$v_{ij} = w_j a_{ij}. \tag{25}$$

Step 12. Positive and negative ideal solution values are calculated. The maximum value of each column is called the positive ideal solution $\left(v_j^+\right)$, and the minimum value is called the negative ideal solution $\left(v_j^-\right)$.

$$v_j^+ = (\, v_1, \ v_2, \ \ldots, \ v_j) \text{ maximum values}, \quad v_j^- = (\, v_1, \ v_2, \ \ldots, \ v_j) \text{ minimum values}.$$

Step 13. Positive and negative distance values are determined.

$$S^+ = \sqrt{\sum_{j=1}^{m} \left( v_{ij} - v_j^+ \right)^2} \qquad (26)$$

$$S^- = \sqrt{\sum_{j=1}^{m} \left( v_{ij} - v_j^- \right)^2}. \qquad (27)$$

Step 14. The closeness coefficient is calculated according to the ideal solution. Then the closeness coefficient values are normalized.

$$C_i = \frac{S_i^-}{S_i^- + S_i^+} \qquad (28)$$

$$C_i^* = \frac{C_i}{\sum_{i=1}^{n} C_i}. \qquad (29)$$

Step 15. Ranked from largest to smallest according to the normalised closeness coefficient.

## APPLICATION

The approach proposed in this chapter is applied to a plastic injection moulding machine risk assessment problem in a small household appliance manufacturing company. In this study, criteria are considered risk parameters, and alternatives are considered hazards.

### Hierarchy of evaluation criteria, alternatives and decision makers

Risk assessment is carried out by systematically analysing the identified risk parameters and hazards. Risk parameters and hazards in the hierarchical structure in Fig. 3 are determined in line with the literature review, process analysis and expert opinions. Risk parameters are determined to assess the health and safety risks of workers working on plastic injection moulding machines. Probability (P1), severity (P2), frequency (P3), detectability (P4), human error (P5), machine error (P6) and existing safety measures (P7) are used as risk parameters. A total of nine potential hazards that may arise from the use of plastic injection moulding machines have been identified. These hazards and their descriptions are presented in Table 4.

Decision-makers are determined according to their experience in the relevant field (at least 5 years), their level of education (at least high school) and their titles (employer, occupational safety specialist, workplace physician, production manager, employee representative, experienced employee, *etc.*). Details of the decision-makers are given in Table 5.

Since the decision-making methods used in the risk assessment process take into account subjective judgements, it is aimed to create a decision-making group that can fully comprehend the meaning and importance of the assessment to be made. Accordingly, different SDMGs are determined for each evaluation area. SDMGs are determined in line with the common opinions of the study team, academics and field experts. SDMGs are determined separately according to each assessment area. Details of the SDMGs and evaluation areas are presented in Table 6.

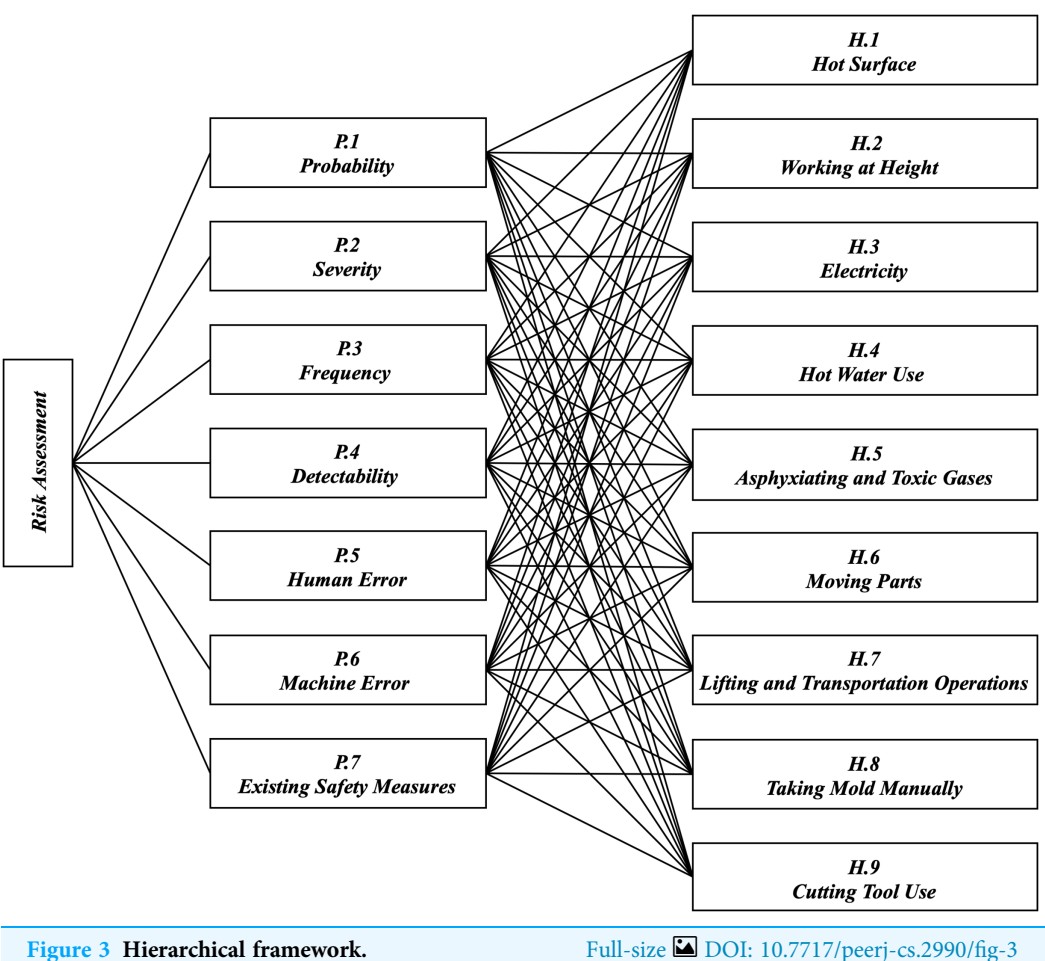

**Figure 3 Hierarchical framework.**               

---

**Table 4 Hazards of plastic injection moulding machine.**

| Hazards code and hazards | Description of hazards | References |
|---|---|---|
| H1: Hot Surface | Plastic injection moulding machine is used to melt and inject plastic materials at high temperature. During this process, many components and parts of the machine are exposed to high temperatures. Mould processing units, pipes and heaters are particularly at risk of hot surfaces. | *Aneziris, Papazoglou & Doudakmani (2010)*, *Pacana (2017)*, *Pathak (2019)*, *Rahim & Raman (2017)* |
| H2: Working at Height | Since plastic injection moulding machines are large machines, workers working on these machines work at height during maintenance, cleaning, *etc.* of raw material silos. | *Pacana (2017)*, *Sanyaolu (2022)* |
| H3: Electricity | These machines pose an electrical hazard because they are electrically powered, made of metal material that conducts electricity and have an electrical panel on the machine. | *Aneziris, Papazoglou & Doudakmani (2010)*, *Pacana (2017)*, *Pathak (2019)* |
| H4: Hot Water Use | During the plastic injection moulding process, hot water is kept up to the machine with the help of pipes to keep the plastic in a molten state until the plastic material is melted and poured into the mould. This hot water is transported with the help of pipes and the pipe bursts and sprays during the transportation of pressurized hot water. | (Expert opinion) |

(Continued)

| Hazards code and hazards | Description of hazards | References |
|---|---|---|
| H5: Asphyxiating and Toxic Gases | Asphyxiating and toxic gases released because of inadequate ventilation and the application of heat to the raw materials and additives mixed into the plastic. | *Aneziris, Papazoglou & Doudakmani (2010)* |
| H6: Moving Parts | Movement of moulds in the injection moulding department and lack of protective covers for this area. | *Pacana (2017)*, *Pathak (2019)*, *Rahim & Raman (2017)*, *Sanyaolu (2022)*, *Chinniah (2015)* |
| H7: Lifting and Transportation Operations | Lifting the moulds with overhead cranes during mould change, failure to properly connect all connections and components during mould change, handle, eyebolt control, standing under the material during transportation, damaged crane ropes, failure to perform crane controls, *etc*. | *Aneziris, Papazoglou & Doudakmani (2010)*, *Pacana (2017)*, *Sanyaolu (2022)* |
| H8: Taking Mold Manually | Hand jamming during manual removal of moulds between moving mould plates, burns due to hot mould plates of moulds. | *Chinniah (2015)* |
| H9: Cutting Tool Use | Cutting the hand while cutting the burrs of the final products coming out of the injection moulding machine with the help of a utility knife. | *Chinniah, Aucourt & Bourbonnière (2017)* |

**Table 5  Decision makers.**

| Decision makers ID | Decision makers | Occupation | Experience (year) | Education |
|---|---|---|---|---|
| DM1 | Occupational Safety Specialist | Occupational Safety Specialist (Class B) | 8 | Master's Degree |
| DM2 | Production Manager | Mechanical Engineer | 25 | License |
| DM3 | Academician | Professor | 32 | PhD |
| DM4 | Workplace Physician | Workplace Physician | 7 | License |
| DM5 | Employee Representative | Assistant Production Manager | 8 | License |
| DM6 | Employee | Employee | 8 | License |
| DM7 | Employer | Human Resources Specialist | 14 | License |

**Table 6  Sub-decision maker groups.**

| Sub-decision-maker group | Decision makers ID | Evaluation criteria |
|---|---|---|
| SDMG1 | DM1, DM2, DM3 | Evaluation of risk parameters |
| SDMG2 | DM1, DM2, DM3, DM4, DM5, DM6, DM7 | Probability, Severity |
| SDMG3 | DM1, DM2, DM5, DM6, DM7 | Frequency |
| SDMG4 | DM1, DM2, DM3, DM5, DM6, DM7 | Human error, Machine error, Existing safety measures |
| SDMG5 | DM1, DM2, DM3, DM7 | Detectability |

## Determining the weights of the risk parameters using IVFF-AHP

In this phase, the weight of the risk parameters is calculated according to the procedures from Step 1 to Step 10 of the IVFF-AHP method. Decision-makers use the linguistic terms and scales presented in Table 2 when assessing risk parameters. An example of calculating the weight of risk parameters is given below.

Step 1: After determining the risk parameters, a hierarchical framework is created as shown in Fig. 3. To evaluate the risk parameters, the occupational safety specialist (DM1),

the production manager (DM2), academician (DM3) in the 1st sub-decision maker group (SDMG1) are determined as decision makers.

Step 2: Decision-makers are evaluated according to the linguistic terms given in Table 3. The decision makers determined for the risk parameters are evaluated by the working team, academics, and practitioners as occupational safety specialist (VS), production manager (MS) and academician (VS) according to the linguistic terms in Table 3. Linguistic terms are converted into FFs, and the weight of decision-makers is determined by Eq. (13). For example, the weight of the Occupational safety specialist (DM1) decision-maker is determined as follows:

$$DM_1 = (\mu_1, v_1) = (0.75, \ 0.30);$$

$$\psi_1 = \frac{1 + \mu_1^3 - v_1^3}{\sum_{t=1}^{3} (1 + \mu_t^3 - v_t^3)}$$

$$= \frac{1 + (0.75)^3 - (0.30)^3}{\left(1 + (0.75)^3 - (0.30)^3\right) + \left(1 + (0.55)^3 - (0.50)^3\right) + \left(1 + (0.75)^3 - (0.30)^3\right)}$$

$$= 0.3641.$$

Step 3: A pairwise comparison matrix of risk parameters is created. Decision-makers assess risk parameters using the linguistic terms in Table 2. The pairwise comparison matrix of risk parameters is given in File S12. These linguistic terms are converted into IVFFSs given in Table 2.

Step 4: The CR of each pairwise comparison matrix is calculated. The CR of the matrix is determined according to Satty's classical consistency procedure. For example, the CR value of the pairwise comparison matrix of the risk parameters evaluated by the occupational safety specialist (DM1) is calculated as follows. It is obtained as $\lambda\_max = 7.042$ and $n = 7$ belonging to the pairwise comparison matrix. According to Eq. (15), CI = ($\lambda\_max$-n)/(n-1) = (7.042 − 7)/( 7−1) = 0.007. To obtain the CR value, the RI value in the equation is determined as (RI = 1.32) for $n = 7$. It is then obtained as CR = CI/RI = 0.007/1.32 = 0.005. The CR ratios for each matrix are given in the left column of the tables.

Step 5: The decision-makers' evaluations are combined using the IVFFWA operator given in Eq. (17). For example, the evaluations Z12 (SLI) of DM1, Z12 (LI) of DM2, Z12 (LI) of DM3 are combined as follows using the IVFFWA operator. These linguistic terms are converted into IVFFSs in Table 2. The linguistic terms in the pairwise comparison matrix of each decision maker are converted into IVFF numbers in the form of $A_{12}^1 = ([0.35, \ 0.4], [0.6, 0.65])$, $A_{12}^2 = ([0.2, 0.3], [0.7, 0.8])$, $A_{12}^3 = ([0.35, 0.4], [0.6, 0.65])$. Then, the IVFFWA operator operations in Eq. (17) are applied.

IVFFWA $(\tilde{\mathcal{F}}_1, \tilde{\mathcal{F}}_2, \ldots, \tilde{\mathcal{F}}_n)$

$$= \left( \begin{bmatrix} \sqrt[3]{\left(1 - \left(1 - (0.35)^3\right)^{0.3641} * \left(1 - (0.35)^3\right)^{0.2718} * \left(1 - (0.35)^3\right)^{0.3641}\right)}, \\ \sqrt[3]{\left(1 - \left(1 - (0.4)^3\right)^{0.3641} * \left(1 - (0.3)^3\right)^{0.2718} * \left(1 - (0.4)^3\right)^{0.3641}\right)} \end{bmatrix} \right.$$
$$\left. \times \left[ (0.6)^{0.3641} * (0.7)^{0.2718} * (0.6)^{0.3641}, (0.65)^{0.3641} * (0.65)^{0.2718} * (0.65)^{0.3641} \right] \right).$$

IVFFWA $(\tilde{\mathcal{F}}_1, \tilde{\mathcal{F}}_2, \tilde{\mathcal{F}}_3) = ([0.35, \ 0.37], [0.62, 0.65])$ is obtained.

Step 6: The separation between the upper and lower points in the calculation matrix is calculated according to Eqs. (18) and (19). The separation value between the upper and lower points $D = (d_{12})_{7 \times 7}$ evaluated in step 5 is calculated as follows.

$$d_{12}^L = (0.37)^3 - (0.62)^3 = -0.1877$$
$$d_{12}^U = (0.35)^3 - (0.65)^3 = -0.2318$$

Step 7: The interval generative matrix $S = (s_{12})_{7 \times 7}$ is calculated according to Eqs. (20) and (21).

$$s_{12}^L = \sqrt[3]{1000^{d_{12}^L}} = \sqrt[3]{1000^{-0.1877}} = 0.6491$$
$$s_{12}^U = \sqrt[3]{1000^{d_{12}^U}} = \sqrt[3]{1000^{-0.2318}} = 0.5864.$$

Step 8. The uncertainty value $T = (t_{12})_{7 \times 7}$ of the $z_{12}$ is obtained.

$$t_{12} = 1 - (0.35^3 - 0.37^3) - (0.62^3 - 0.65^3) = 1.0441.$$

Step 9. The uncertainty levels are multiplied with $S = (s_{12})_{7 \times 7}$ matrix to find the matrix of unnormalized weights $R = (r_{12})_{7 \times 7}$ by Eq. (23).

$$r_{12} = \left(\frac{s_{12}^L + s_{12}^U}{2}\right) t_{ij} = \left(\frac{0.6491 + 0.5864}{2}\right) * 1.0441 = 0.6450.$$

Step 10. The computation of the normalised priority weights $w_i$ is as below.

$$w_1 = \frac{\sum_{j=1}^m r_{ij}}{\sum_{i=1}^m \sum_{j=1}^m r_{ij}} = 0.1445.$$

The weights of the risk parameters are determined according to the IVFF-AHP calculation steps in the example above. The weights of the decision makers, linguistic assessments of the risk parameters, and consistency ratios are presented in File S12. The consistency ratio of the pairwise comparison matrices is calculated using Excel during the evaluations. The MATLAB code of the calculations of the IVFF-AHP method is given in the File S2. All risk parameters according to highest weight are the existing safety measures (0.1987), detectability (0.1762), machine error (0.1573), human error (0.1558), probability (0.1445), frequency (0.1195) and severity (0.0481), respectively.

## Determination of hazard scores using IVFF-AHP

In this phase, the hazard scores are calculated according to the procedures from Step 1 to Step 10 of the IVFF-AHP method. Decision-makers use the linguistic terms and scales presented in Table 2 when assessing hazards. Pairwise comparison matrices of hazards are created according to each risk parameter. For example, the calculation of the scores of the hazards according to the detectability parameter is given below.

Step 1: After determining the risk parameters and hazards, a hierarchical framework is created as shown in Fig. 3. To evaluate the hazards according to the detectability parameter, the occupational safety specialist (DM1), the production manager (DM2), the academician (DM3) and the employer (DM7), who are in the 5th sub-decision maker group (SDMG5), are determined as decision makers.

Step 2: Decision makers are evaluated according to the linguistic terms given in Table 3. The decision-makers determined for the detectability parameter are evaluated by the working team, academics, and practitioners as occupational safety specialist (AS),

production manager (AS), academician (AS) and employer (MS) according to the linguistic terms in Table 3. Linguistic terms are converted into FFs, and the weight of decision-makers is determined by Eq. (13). For example, the weight of the occupational safety specialist decision-maker (DM1) is determined as follows:

$$DM_1 = (\mu_1, \, v_1) = (0.95, \, 0.10);$$

$$\psi_1 = \frac{1 + \mu_1^3 - v_1^3}{\sum_{t=1}^{4}(1 + \mu_t^3 - v_t^3)}$$

$$= \frac{1 + (0.95)^3 - (0.10)^3}{\left(1 + (0.95)^3 - (0.10)^3\right) + \left(1 + (0.95)^3 - (0.10)^3\right) + \left(1 + (0.95)^3 - (0.10)^3\right) + \left(1 + (0.55)^3 - (0.50)^3\right)}$$

$$= 0.2808.$$

Step 3: A pairwise comparison matrix of hazards is created. Decision-makers assess hazards based on the detectability parameter using the linguistic terms in Table 2. The pairwise comparison matrix of hazards according to the detectability parameter is given in File S13. These linguistic terms are converted into IVFFSs given in Table 2.

Step 4: The CR of each pairwise comparison matrix is calculated. The CR of the matrix is determined according to Satty's classical consistency procedure. For example, the CR value of the pairwise comparison matrix of hazards assessed by the occupational safety specialist (DM1) is calculated as follows. It is obtained as $\lambda\_max = 10.114$ and $n = 9$ belonging to the pairwise comparison matrix. According to Eq. (15), CI = ($\lambda\_max$-n)/(n-1) = (10.114 − 9)/(9 − 1) = 0.139. To obtain the CR value, the RI value in the equation is determined as (RI = 1.45) for $n = 9$. It is then obtained as CR = CI/RI = 0.139/1.45 = 0.096. The CR ratios for each matrix are given in the left column of the tables.

Step 5: The decision-makers' evaluations are combined using the IVFFWA operator given in Eq. (17). For example, the evaluations Z12 (SLI) of DM1, Z12 (SLI) of DM2, Z12 (LI) of DM3 and Z12 (LI) of DM7 are combined as follows using the IVFFWA operator. These linguistic terms are converted into IVFFSs in Table 2. The linguistic terms in the pairwise comparison matrix of each decision maker are converted into IVFF numbers in the form of $A_{12}^1 = ([0.35, 0.4], [0.6, 0.65])$, $A_{12}^2 = ([0.35, 0.4], [0.6, 0.65])$, $A_{12}^3 = ([0.2, 0.3], [0.7, 0.8])$ ve $A_{12}^7 = ([0.2, 0.3], [0.7, 0.8])$. Then, the IVFFWA operator operations in Eq. (17) are applied. IVFFWA $(\tilde{\mathcal{F}}_1, \tilde{\mathcal{F}}_2, \ldots, \tilde{\mathcal{F}}_n)$

$$= \left(\left[\begin{array}{l} \sqrt[3]{\left(1 - \left(1 - (0.35)^3\right)^{0.2808} * \left(1 - (0.35)^3\right)^{0.2808} * \left(1 - (0.2)^3\right)^{0.2808}\right) * \left(1 - (0.2)^3\right)^{0.1575}}, \\ \sqrt[3]{\left(1 - \left(1 - (0.4)^3\right)^{0.2808} * \left(1 - (0.4)^3\right)^{0.2808} * \left(1 - (0.3)^3\right)^{0.2808}\right) * \left(1 - (0.3)^3\right)^{0.1575}} \end{array}\right]\right.$$
$$\left. \times \left[(0.6)^{0.2808} * (0.6)^{0.2808} * (0.7)^{0.2808} * (0.7)^{0.1575}, (0.65)^{0.2808} * (0.65)^{0.2808} * (0.8)^{0.2808} * (0.8)^{0.1575}\right]\right)$$

IVFFWA $(\tilde{\mathcal{F}}_1, \tilde{\mathcal{F}}_2, \tilde{\mathcal{F}}_3) = ([0.29, 0.37], [0.72, 0.70]).$

Step 6: The difference between the upper and lower points in the calculation matrix is calculated according to Eqs. (18) and (19). The difference value between the upper and lower points $D = (d_{12})_{9 \times 9}$ evaluated in step 5 is calculated as follows.

$$d_{12}^L = (0.37)^3 - (0.72)^3 = -0.3226$$

$d^U_{12} = (0.29)^3 - (0.70)^3$ = -0.3186.

Step 7: The interval generative matrix $S = (s_{12})_{9\times9}$ is calculated according to Eqs. (20) and (21).

$$s^L_{12} = \sqrt[3]{1000^{d^L_{12}}} = \sqrt[3]{1000^{-0.3226}} = 0.7777$$

$$s^U_{12} = \sqrt[3]{1000^{d^U_{12}}} = \sqrt[3]{1000^{-0.3186}} = 0.7790$$

Step 8. The uncertainty value $T = (t_{12})_{9\times9}$ of the $z_{12}$ is obtained.

$$t_{12} = 1 - ((0.29)^3 - (0.37)^3) - ((0.72)^3 - (0.70)^3) = 0.9960.$$

Step 9. The uncertainty levels are multiplied with $S = (s_{12})_{9\times9}$ matrix to find the matrix of unnoted weights $R = (r_{12})_{9\times9}$ by Eq. (23).

$$r_{ij} = \left(\frac{s^L_{12} + s^U_{12}}{2}\right)t_{12} = \left(\frac{0.7777 + 0.7790}{2}\right) * 0.9960 = 0.7764.$$

Step 10. The computation of the normalised priority weights $w_i$ is as below.

$$w_1 = \frac{\sum_{j=1}^9 r_{ij}}{\sum_{i=1}^9 \sum_{j=1}^9 r_{ij}} = 0.0798.$$

The scores of the hazards are determined according to the IVFFAHP calculation steps in the example given above. The weights of the decision makers, linguistic assessments of hazards assessed according to the parameters of probability, severity, frequency, detectability, human error, machine error, and existing safety precautions, and consistency ratios are presented in the from Files S13 to S19. The consistency ratio of the pairwise comparison matrices is calculated using Excel during the evaluations. The MATLAB codes of the calculations of the IVFF-AHP method are given in the from Files S3 to S9. The hazard scores obtained according to each risk parameter are given in Table 7.

## Determination of risk coefficient values using TOPSIS

In this phase, after the hazard scores are obtained with the IVFFAHP method, the procedures from Step 11 to Step 15 of the TOPSIS method are applied to obtain the risk coefficient values. Hazard scores and risk parameter weights obtained by the IVFFAHP method are directly transferred to the weighted decision matrix. In this study, detectability and existing safety measures are defined as cost criteria, while other risk parameters are defined as benefit criteria. The risk coefficient values of the hazards are obtained by the TOPSIS method. According to the risk coefficient values, hazards are ranked starting from the highest risk hazard. An example of the calculation of the ranking of hazards is given below.

Step 11. A weighted decision matrix $V = (v_{11})_{9\times7}$ is created using the risk parameters and hazard scores. Where $w_1$ is the risk parameters weight. $a_{11}$ is the evaluation of the 1. hazard according to the 1. risk parameter.

$$v_{11} = w_1 a_{11} = 0.1445 * 0.0791 = 0.0114$$

Step 12. Positive and negative ideal solution values are calculated. The maximum value of each column is called the positive ideal solution $\left(v^+_j\right)$, and the minimum value is called the negative ideal solution $\left(v^-_j\right)$.

**Table 7 Decision matrix.**

|  |  | Parameter weights | | | | | | |
| --- | --- | --- | --- | --- | --- | --- | --- | --- |
|  |  | 0.1445 | 0.0481 | 0.1195 | 0.1762 | 0.1558 | 0.1573 | 0.1987 |
|  |  | P1 | P2 | P3 | P4 | P5 | P6 | P7 |
| Hazard scores | H1 | 0.0791 | 0.1827 | 0.0865 | 0.0798 | 0.0868 | 0.1622 | 0.1493 |
|  | H2 | 0.1438 | 0.0490 | 0.1561 | 0.0533 | 0.0644 | 0.1096 | 0.1276 |
|  | H3 | 0.1678 | 0.0286 | 0.1831 | 0.1572 | 0.1135 | 0.1427 | 0.0733 |
|  | H4 | 0.2059 | 0.1824 | 0.2007 | 0.1686 | 0.1858 | 0.0922 | 0.1562 |
|  | H5 | 0.1829 | 0.1157 | 0.1183 | 0.1642 | 0.1819 | 0.0758 | 0.0763 |
|  | H6 | 0.0402 | 0.0653 | 0.0542 | 0.1359 | 0.1064 | 0.0572 | 0.0421 |
|  | H7 | 0.0816 | 0.0586 | 0.1117 | 0.1171 | 0.1536 | 0.0102 | 0.0619 |
|  | H8 | 0.0589 | 0.1427 | 0.0430 | 0.0831 | 0.0836 | 0.1310 | 0.1395 |
|  | H9 | 0.0398 | 0.1751 | 0.0463 | 0.0408 | 0.0240 | 0.2192 | 0.1737 |

$$v_j^+ = (v_1, v_2, \ldots, v_j) = (0.0298, 0.0088, 0.0240, 0.0072, 0.0289, 0.0345, 0.0084)$$
$$v_j^- = (v_1, v_2, \ldots, v_j) = (0.0058, 0.0014, 0.0051, 0.0297, 0.0037, 0.0016, 0.0345).$$

Step 13. Positive and negative distance values are determined.

$$S^+ = \sqrt{\sum_{j=1}^{7} \left(v_{ij} - v_j^+\right)^2}$$

$$= \sqrt{(v_{11} - v_1^+)^2 + (v_{12} - v_2^+)^2 + (v_{13} - v_3^+)^2 + (v_{14} - v_4^+)^2 + (v_{15} - v_5^+)^2 + (v_{16} - v_6^+)^2 + (v_{17} - v_7^+)^2}$$

$$= \sqrt{\begin{array}{l}(0.0003 - 0.0298)^2 + (0.0000 - 0.0088)^2 + (0.0002 - 0.0240)^2 + (0.0002 - 0.0297)^2 + (0.0002 - 0.0289)^2 \\ + (0.0001 - 0.0345)^2 + (0.0000 - 0.0345)^2\end{array}}$$

$$= 0.0366$$

$$S^- = \sqrt{\sum_{j=1}^{n} \left(v_{ij} - v_j^-\right)^2}$$

$$= \sqrt{(v_{11} - v_1^-)^2 + (v_{12} - v_2^-)^2 + (v_{13} - v_3^-)^2 + (v_{14} - v_4^-)^2 + (v_{15} - v_5^-)^2 + (v_{16} - v_6^-)^2 + (v_{17} - v_7^-)^2}$$

$$= \sqrt{\begin{array}{l}(0.0000 - 0.0058)^2 + (0.0001 - 0.0014)^2 + (0.0000 - 0.0051)^2 + (0.0000 - 0.0072)^2 + (0.0001 - 0.0037)^2 \\ + (0.0006 - 0.0016)^2 + (0.0005 - 0.0084)^2\end{array}}$$

$$= 0.0324.$$

Step 14. The closeness coefficient is calculated according to the ideal solution. Then the closeness coefficient values are normalized.

$$C_1 = \frac{S_1^-}{S_1^- + S_1^+} = \frac{0.0324}{0.0324 + 0.0366} = 0.4694$$

$$C_1^* = \frac{C_1}{\sum_{i=1}^{7} C_i} = \frac{0.4694}{4.3474} = 0.1080.$$

Biderci and Guneri (2025), *PeerJ Comput. Sci.*, DOI 10.7717/peerj-cs.2990

**Table 8 Different cases of changing risk parameters and their descriptions.**

| Cases | Descriptions |
|---|---|
| Current case | Weights obtained with the IVFF-AHP method |
| C1 | All parameters have equal weights (0.1429) |
| C2 | The weight of the most important parameter is fixed and is the highest (WP7 = 0.1987). The others are the same (0.1335) |
| C3 | The weight of the least important parameter is fixed and the highest (WP2 = 0.1987). The others are the same (0.1335) |
| C4 | The weights of the first three most important parameters (WP4, WP5 ve WP7) are fixed at 0.1335. The others are the same (0.1499) |
| C5 | The weights of the first three least important parameters (WP1, WP2 ve WP3) are fixed at 0.1335. The others are the same (0.1499) |

Step 15. Ranked from largest to smallest according to the normalized closeness coefficient. In this study, normalized closeness is expressed as risk coefficients of hazards. The MATLAB code for the calculations of the TOPSIS method is given in the File S10. Accordingly, the ranking of the hazards with the highest risk coefficient value is as follows: Electricity (0.1559) > Asphyxiating and Toxic Gases (0.1432) > Hot Water Use (0.1379) > Lifting and Transportation Operations (0.1027) > Moving Parts (0.1016) > Hot Surface (0.0959) > Working at Height (0.0941) > Cutting Tool Use (0.0879) > Taking Mould Manually (0.0808).

## Sensitivity analysis

A sensitivity analysis is performed to assess how decision-making processes and outcomes react to uncertainties and changes. A sensitivity analysis in this study is performed by changing the risk parameter weights. With sensitivity analysis, the scenarios in the ranking results are evaluated depending on the changes in the weights of the risk parameters. Scenarios for changes in risk parameter weights and their explanations are presented in Table 8. The scenario results and rankings are shown in Table 9. In Table 9, it can be observed that when the risk parameter weights change, the risk coefficient values of the results change. Therefore, the proposed approach is sensitive to the weights of risk parameters. Sensitivity analysis of the proposed methodology shows that changing the risk parameter weights can have an impact on the final results, as expected. However, despite the variation of the risk coefficient values, H3, H4 and H5 are ranked as the most important hazards depending on all situations and are less susceptible to different weightings. This consistency indicates the robustness of the riskiest hazards. In Cases 4 and 5, the order of the nine hazards remains the same, except for minor differences. Differences in rankings for different risk parameter weights indicate the sensitivity of the proposed model to changes. Thus, the applicability and reliability of the proposed model to changing conditions are proven.

## Comparative analysis

In this section, the risk coefficient ranking obtained by the IVFFAHP-TOPSIS method is compared with the ranking results of the simple additive weighting (SAW), multi-objective optimization by ratio analysis (MOORA) and evaluation based on distance from average solution (EDAS) methods integrated into the IVFFAHP method. The risk coefficient of the hazards obtained through the software and rankings are given in Table 10. In addition,

**Table 9  The scenario results of different cases.**

|  | Current case | Case 1 | Case 2 | Case 3 | Case 4 | Case 5 |
|---|---|---|---|---|---|---|
| H1 | 0.0959 (6) | 0.1189 (4) | 0.1121 (4) | **0.1289 (2)** | 0.1206 (4) | 0.1191 (4) |
| H2 | 0.0941 (7) | 0.1115 (5) | 0.1082 (5) | 0.1021 (7) | 0.1121 (6) | 0.1108 (6) |
| H3 | **0.1559 (1)** | **0.1249 (3)** | **0.1281 (3)** | 0.1114 (5) | **0.1262 (2)** | **0.1246 (3)** |
| H4 | **0.1379 (3)** | **0.1400 (1)** | **0.1311 (1)** | **0.1455 (1)** | **0.1430 (1)** | **0.1353 (1)** |
| H5 | **0.1432 (2)** | **0.1265 (2)** | **0.1292 (2)** | **0.1262 (3)** | **0.1260 (3)** | **0.1248 (2)** |
| H6 | 0.1016 (5) | 0.0794 (9) | 0.0937 (8) | 0.0765 (9) | 0.0753 (9) | 0.0824 (9) |
| H7 | 0.1027 (4) | 0.0918 (8) | 0.1002 (7) | 0.0868 (8) | 0.0875 (8) | 0.0930 (8) |
| H8 | 0.0808 (9) | 0.0963 (7) | 0.0929 (9) | 0.1044 (6) | 0.0965 (7) | 0.0978 (7) |
| H9 | 0.0879 (8) | 0.1108 (6) | 0.1045 (6) | 0.1181 (4) | 0.1128 (5) | 0.1123 (5) |

Note:
  Ranking results are given in parentheses. The three highest scores are indicated in bold.

**Table 10  Comparative analysis results.**

|  | TOPSIS | SAW | MOORA | EDAS |
|---|---|---|---|---|
| H1 | 0.0959 (6) | 0.1073 (4) | 0.1022 (5) | 0.1055 (5) |
| H2 | 0.0941 (7) | 0.1058 (5) | 0.1390 (4) | 0.1328 (4) |
| H3 | **0.1559 (1)** | **0.1347 (3)** | **0.1851 (1)** | **0.1584 (1)** |
| H4 | **0.1379 (3)** | **0.1798 (1)** | **0.1820 (2)** | **0.1430 (3)** |
| H5 | **0.1432 (2)** | **0.1360 (2)** | **0.1726 (3)** | **0.1480 (2)** |
| H6 | 0.1016 (5) | 0.0648 (9) | 0.0324 (9) | 0.0657 (8) |
| H7 | 0.1027 (4) | 0.0858 (7) | 0.0831 (6) | 0.0914 (7) |
| H8 | 0.0808 (9) | 0.0837 (8) | 0.0428 (8) | 0.0607 (9) |
| H9 | 0.0879 (8) | 0.1020 (6) | 0.0608 (7) | 0.0946 (6) |
| r |  | 0.7381 | 0.8422 | 0.8159 |

Note:
  Ranking results are given in parentheses. The three highest scores are indicated in bold.

pearson correlation coefficient values are calculated to examine the relationship between the ranking results of different MCDM methods, and these values are given in Table 10. When the rankings are analysed, it is seen that the highest-risk and lowest-risk hazards are in similar rankings. In TOPSIS, MOORA and EDA S methods, the hazard with the highest risk is determined as electricity (H3). However, the lowest risk hazard differs according to the methods. As can be seen in Table 10, the order of the riskiest hazards is like each other, although there are slight differences in the risk coefficient results between the methods. Similar results obtained because of applying the proposed method to different MCDM methods show the effectiveness and consistency of the proposed method. When the pearson corelation coefficients are examined, it is shown that the highest similarity rate is between the TOPSIS method and the MOORA method (r = 0.8422). There is a strong positive correlation between all MCDM methods, and it is concluded that IVFFAHP-TOPSIS shows a high degree of positive correlation with the results of other methods. The results of the pearson correlation test confirm that the ranking results of the TOPSIS method are consistent with the ranking results of other methods.

## DISCUSSION

This study presents a new decision-making approach based on the selection of decision makers according to evaluation criteria. The risk parameters used for the risk assessment study are determined as evaluation criteria. Accordingly, SDMGs for each evaluation criterion are formed. The SDMGs approach, created according to evaluation criteria, offers a more flexible and dynamic structure than existing approaches. This decision-making approach aims to use the expertise and knowledge of decision-makers more effectively. The study showed that the determination of decision-makers separately according to the evaluated criteria is applicable to the decision-making process. The approach presented in this study offers an innovative model and adds a new dimension to decision-making processes. There is no study in the literature on this approach, and this new decision-making approach makes a significant contribution to the literature.

This decision-making approach is applied to the risk assessment, as it involves the evaluation of different criteria. When the risk assessment, risk parameters are determined as criteria during the application of MCDM methods. In addition to the classical risk parameters such as probability, severity, frequency, and detectability, new parameters such as human error, machine error, and existing safety measures are used in the risk assessment study. These additional parameters provide a more comprehensive assessment of risks by focusing on the direct effects of human error and machine error on risk and the role of existing safety measures. In particular, the identification of existing safety measures as the most important risk parameter emphasises the central role of this factor in risk management. This finding clearly emphasises the importance of investments in safety measures and underlines the critical role of safety measures compared to previous studies. In addition, it is seen that machine error and human error parameters are also important risk parameters according to the results of the study. These results show that these risk parameters included in the risk assessment contribute to a better understanding of different aspects of risks and the development of a more comprehensive risk assessment strategy. These parameters, which are used for the first time in the literature, provide significant advantages in risk management by providing a broader perspective compared to traditional methods.

The integration of AHP and TOPSIS methods into the IVFFS environment provides an important contribution to a consideration of risks and uncertainties more comprehensively in the risk assessment process. Due to its advantages over other MCDM methods, the AHP method is used to weight the risk parameters and determine the hazard scores, and the TOPSIS method is used to rank the hazards. A holistic and systematic approach to risk assessment has been achieved by integrating these two methods. Risk parameter weights, scores, and ranking of hazards may change over time. The codes of these methods, which are written in the MATLAB_R2024a software program, enable them to adapt quickly to changing conditions and give a dynamic structure to the evaluation process. In this respect, transferring MCDM methods to the software environment both accelerates the decision processes and makes them more systematic. In addition, these methods provide clear and understandable results to decision-makers, reducing

subjectivity and enabling more reliable evaluations. This integration contributes to a more detailed and reliable implementation of the risk assessment process.

The focus of the study on plastic injection moulding machines highlights a gap in the literature where studies on these machines focus on industrial process optimisation rather than occupational safety risks. The application of the risk assessment study to plastic injection moulding machines makes a significant contribution to the understanding of occupational safety risks of this machine. The risks of hazards identified related to plastic injection moulding machines are assessed. According to the assessments, electricity, asphyxiating and toxic gases, and hot water use are determined as the riskiest hazards. These hazards stand out as risks where the risk is particularly likely to occur, workers are most frequently exposed, and human error is high.

Assessments of risk parameters and hazards may change in situations such as changes in working conditions and improvements in safety measures over time. Considering this situation, risk assessment processes should be regularly monitored and updated when necessary. In this way, more up-to-date and reliable results are obtained for risk assessment.

The sensitivity analysis and comparative analysis carried out at the end of the study confirm that the proposed methodology produces consistent and reasonable results. These analyses strengthen the applicability of the methodology and the reliability of the results. Thus, it can be said that the proposed approach supports decision-makers to make informed decisions and provides reliable results.

## CONCLUSIONS

This decision-making approach has been applied in the risk assessment study, which has many different evaluation criteria. A risk assessment study is conducted to assess occupational safety risks in plastic injection moulding machines. In the risk assessment study, probability, severity, frequency, detectability, human error, machine error, and existing safety measures risk parameters are taken into consideration to determine the risk value. Risk parameters are determined as criteria in the risk assessment study. A total of five sub-decision maker groups are formed from seven decision makers (occupational safety specialist, production manager, academician, workplace physician, employee representative, worker, employer) according to different criteria such as probability, severity, frequency, detectability, human error, machine error and existing safety measures used as risk parameters. Evaluations are made by decision-makers in these sub-decision maker groups. Human error, machine error, and existing safety measures are determined as the most important risk parameters. Although these used risk parameters are new in the literature, they are considered important parameters by decision-makers in determining the risk value. The weights of these risk parameters are determined by the AHP method. For risk assessment, hot surface, working at height, electricity, hot water use, asphyxiating and toxic gases, moving parts, lifting and transportation operations, taking mould manually, and cutting tool use hazards of this machine are determined. The scores of these hazards are determined by the AHP method. The TOPSIS method is integrated into the AHP method to determine and rank the risk value of the hazards. The integration of AHP

and TOPSIS methods has been extended to IVFFSs to overcome uncertainties in assessments. The hazards with the highest risk coefficient are determined as electricity, asphyxiating and toxic gases, and hot water use.

By coding the application of the study in the MATLAB_R2024a program, analyses can be easily made for various application areas. In this way, the reusability and development of the application also save time and can be integrated into various application areas.

The sensitivity analysis and comparative analysis confirm that the proposed approach produces consistent and reasonable results and reinforces its validity. These results show that the proposed approach is reliable and applicable in risk assessment processes.

This study has some limitations. The potential hazard used in the study and the number of decision-makers are limitations. Increasing the number of potential hazards can contribute to a more comprehensive risk assessment. Additionally, increasing the number of decision-makers and adding different perspectives can add value to the decision-making process. In addition, there is subjectivity due to biases in the selection of decision-makers and SDMGs. Risk parameters are established taking into account specific conditions, and their adaptability for different systems and areas must be tested. In addition, the developed decision-making approach can be tested for practical applicability to different problems other than risk assessment.

Future studies can address some aspects of this study with different fuzzy-based MCDM methods that can be applied for the weighting of criteria, evaluation and ranking of alternatives. It is possible to use different MCDM methods such as BWM, VIKOR, and COPRAS to obtain comparable results. The proposed decision-making framework can be incorporated into different decision-making processes. Thus, the generalisability of the approach and its applicability to different fields can be tested. It is recommended that this approach be applied in future studies in different industries and application areas in order to expand the scope of application.

## ACKNOWLEDGEMENTS

The authors would like to acknowledge that this article is submitted in partial fulfilment of the requirements for PhD. degree at Yildiz Technical University.

### Funding
The authors received no funding for this work.

### Competing Interests
The authors declare that they have no competing interests.

### Author Contributions
- Hilal Biderci conceived and designed the experiments, performed the experiments, analyzed the data, performed the computation work, prepared figures and/or tables, authored or reviewed drafts of the article, and approved the final draft.

- Ali F. Guneri conceived and designed the experiments, performed the experiments, analyzed the data, performed the computation work, prepared figures and/or tables, authored or reviewed drafts of the article, and approved the final draft.

## Data Availability

The MATLAB codes of the IVFFAHP-TOPSIS methods are available in the Supplemental Files.

These codes were used to obtain the risk parameter weights and hazard scores for each risk parameter using the steps of the IVFFAHP method. The TOPSIS method integrated into the IVFFAHP method was used to obtain the risk coefficient values obtained according to the risk parameter weights and hazard scores.

## Supplemental Information

Supplemental information for this article can be found online at http://dx.doi.org/10.7717/peerj-cs.2990#supplemental-information.

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
