# Peer review of "Risk assessment based on a new decision-making approach with fermatean fuzzy sets"

_PeerJ Computer Science, doi:10.7717/peerj-cs.2990_

## Round 0.1 · original submission · Major Revisions

Based on the three reviewers' comments and suggestions, this manuscript needs major revisions.

Reviewer 1 ·

Basic reporting

English can be improved

Literature references are sufficient and recent

Some figures are poorly prepared and are in bad resolution

Experimental design

no comment

Validity of the findings

no comment

Additional comments

This study applies an existing decision-making framework, the Interval Valued Fermatean Fuzzy Analytic Hierarchy Process combined with Technique for Order Preference by Similarity to Ideal Solution (IVFF-AHP-TOPSIS), to enhance risk assessment for plastic injection molding machines.

The paper tackles an interesting and complex decision problem with a sound methodological basis. Its preliminaries and literature review are well-prepared, and the inclusion of sensitivity analysis strengthens the reliability of the findings. The discussion section is insightful and highlights the importance of incorporating new parameters into risk assessment.

By introducing sub-decision-maker groups for each criterion, the approach offers a dynamic and expertise-driven evaluation structure. While the application of IVFF-AHP-TOPSIS in this context is innovative, the study requires some revision to clarify its contributions, particularly distinguishing its use of existing methodologies from the development of novel concepts or frameworks.

See detailed list of suggested improvements and comments below:

- Please underline the main contribution of the study in the introduction to clarify whether it is a methodological paper or an application-focused one. The current abstract and introduction do not make this distinction clear.
- It is stated that the IVFF-AHP-TOPSIS method was developed by Alkan and Kahraman; this should be explicitly mentioned in the paper to clarify that it is being applied here rather than newly developed.
- Figure 1 is titled "Comparison IFSs, PFSs, and FFSs," but the figure itself contains FFN, PFN, and IFN. This inconsistency should be corrected.
- The quality of Figures 2 and 4 is poor. Please use vector graphics or high-resolution images to improve their readability and presentation.
- In equation (15), lambda_max requires correction for consistency, as it is used as lambdamax in next line. Also, carefully check all other equation to made symbols consistent and explained.
- The study would benefit from a comparison of the proposed approach with other methodologies or an explanation of why such a comparison is not feasible. Additionally, elaborate on the specific advantages of the proposed framework over a simpler AHP + TOPSIS combination to justify its complexity.

Cite this review as

Reviewer 2 ·

Basic reporting

In this paper, the authors developed an MCDM model based on the AHP and TOPSIS methods in Fermatean fuzzy environment. This model was used for the plastic injection molding machine risk assessment study. The model was used for group decision-making.
The paper has certain qualities, but it is necessary to make major changes in order to be accepted for publication.

Experimental design

My comments are as follows:
1. Show application of methods step by step.
2. Show one calculation example for each step.
3. Add the limitations of the model through a separate section.

Validity of the findings

My comments are as follows:
1. Add a comparative analysis. Comparative analysis should prove the quality of the created model.

Additional comments

My comments are as follows:
1. Separate the introduction and literature analysis.
2. The introduction should have given answers about: Practical and Methodological Aims of the Study; Motivation for Conducting Research; The contributions of the study; Organization of the Paper. This should be some subsections. Part of the answer can already be found in the text, and part needs to be add.
3. The literature review is good in terms of the number of papers analyzed. However, the literature review should be divided into three subsections. The first deals with the problem, the second deals with the application of the AHP method, and the third deals with the application of the TOPSIS method. Also, the literature review should have more recent papers (period 2024-2025), such as: Kizielewicz, B., & Sałabun, W. (2024). SITW Method: A New Approach to Re-identifying Multi-criteria Weights in Complex Decision Analysis. Spectrum of Mechanical Engineering and Operational Research, 1(1), 215-226; Biswas, A., Gazi, K. H., Sankar, P. M., & Ghosh, A. (2024). A Decision-Making Framework for Sustainable Highway Restaurant Site Selection: AHP-TOPSIS Approach based on the Fuzzy Numbers. Spectrum of Operational Research, 2(1), 1-26.

4. Explain in detail why the AHP and TOPSIS methods were used to solve the problem in a fuzzy Fermatian environment, why other methods such as FUCOM, VIKOR, MABAC, etc. were not used, and why not some other type of fuzzy numbers.
5. In Figure 2, add sensitivity analysis and comparative analysis.
6. Add future research in the Conclusion.

Cite this review as

Reviewer 3 ·

Basic reporting

This paper studies Risk assessment based on a new decision-making approach with fermatean fuzzy sets. It is interesting. I suggest major revision. Some comments are provided as follows.
The English writing should be polished with help of professionals since there are some typos and grammatical errors.

When mentioning MCDM, IFS and PFS in Introduction, the following related and important references should be commented as follows:
Information Sciences, 666 (2024) 120404; Expert Systems with Applications, 242 (2024) 122456; Applied Soft Computing, 154 (2024) 111374; Information Sciences, 668 (2024) 120526; Journal of Management Analytics, 11(3) (2024) 389-444; Knowledge-Based Systems, 301 (2024) 112300; Expert Systems with Applications, 262 (2025) 125342. It is necessary to make an overall review of literature to grasp the status of research.

The motivations should be elaborated. Why does this paper combine TOPSIS and AHP? What is the advantage of combining TOPSIS and AHP?

In section “Preliminaries on IVFFSs”, all definitions should be marked with citations.

In section “IVFFAHP-TOPSIS method”, please explain the reasonability and rationality on steps 4, 6-10.

Please elaborate the advantage of the proposed IVFFAHP-TOPSIS method with existing other methods.

Please add some solid comparative analyses in section “Discussion” to strengthen the convincingness and credibility of this paper.

In sum, I recommend major revision.

Experimental design

Please add some solid comparative analyses in section “Discussion” to strengthen the convincingness and credibility of this paper.

Validity of the findings

Please add some solid comparative analyses in section “Discussion” to strengthen the convincingness and credibility of this paper.

Additional comments

In sum, I recommend major revision.

Cite this review as

---

## Round 0.2 · Minor Revisions

Editor‘s comments:

1. Please confirm whether the references from these reviewers’ suggestions are really needed. Please cite only necessary references in this article.

2. In the introduction, I read "The authorities who directly affect the decision-making process and make the final decisions are the decision-makers. Therefore, decision-makers who carry out the decision-making process are considered important actors in MCDM problems. Decision-makers are also individuals who understand the importance of their decisions and think about how these decisions are made [1]." In [1], I don't see any content that supports these sentences.

Therefore, this article needs minor revisions.

Reviewer 1 ·

Basic reporting

no comment

Experimental design

no comment

Validity of the findings

no comment

Additional comments

The authors improved the paper sufficiently and I think it can be accepted in the current form.

Cite this review as

Reviewer 2 ·

Basic reporting

No comment.

Experimental design

No comment.

Validity of the findings

No comment.

Additional comments

All the reviewers' comments have been addressed carefully and sufficiently. The revisions are rational from my point of view. I think the current version of the paper can be accepted.

Cite this review as

Reviewer 3 ·

Basic reporting

It can be accepted now.

Experimental design

It can be accepted now.

Validity of the findings

It can be accepted now.

Additional comments

It can be accepted now.

Cite this review as

---

## Round 0.3 · Minor Revisions

Dear authors,

In this article, the authors cite 153 references, which is too many. In this case, you only need to cite references related to Fuzzy Sets and Fermatean Fuzzy Sets, and reduce citations to other references.

---

## Round 0.4 · accepted · Accept

Based on the revised conversion (V3), this article can be accepted for publication.